# Cooperative Multi-Sensor Tracking of Vulnerable Road Users in the Presence of Missing Detections

**DOI:** 10.3390/s20174817

**Published:** 2020-08-26

**Authors:** Martin Dimitrievski, David Van Hamme, Peter Veelaert, Wilfried Philips

**Affiliations:** TELIN-IPI, Ghent University—Imec, St-Pietersnieuwstraat 41, B-9000 Gent, Belgium; david.vanhamme@ugent.be (D.V.H.); peter.veelaert@ugent.be (P.V.); wilfried.philips@ugent.be (W.P.)

**Keywords:** multi-object tracking, switching observation model, multiple imputations, particle filter, cooperative sensor fusion, people tracking, radar, camera

## Abstract

This paper presents a vulnerable road user (VRU) tracking algorithm capable of handling noisy and missing detections from heterogeneous sensors. We propose a cooperative fusion algorithm for matching and reinforcing of radar and camera detections using their proximity and positional uncertainty. The belief in the existence and position of objects is then maximized by temporal integration of fused detections by a multi-object tracker. By switching between observation models, the tracker adapts to the detection noise characteristics making it robust to individual sensor failures. The main novelty of this paper is an improved imputation sampling function for updating the state when detections are missing. The proposed function uses a likelihood without association that is conditioned on the sensor information instead of the sensor model. The benefits of the proposed solution are two-fold: firstly, particle updates become computationally tractable and secondly, the problem of imputing samples from a state which is predicted without an associated detection is bypassed. Experimental evaluation shows a significant improvement in both detection and tracking performance over multiple control algorithms. In low light situations, the cooperative fusion outperforms intermediate fusion by as much as 30%, while increases in tracking performance are most significant in complex traffic scenes.

## 1. Introduction

Perception systems for intelligent vehicles have the difficult task of improving on the human capacity for scene understanding. Contemporary autonomous prototypes tackle this challenge using various sensing and processing technologies including cameras, lidar, stereo, radar, and so forth, along with algorithms for object detection, semantical segmentation, temporal tracking, collision avoidance to name a few. With the advent of deep learning, the capability of individual algorithms is becoming ever higher. At the same time, sensor precision is steadily increasing while the price of hardware is going down as demand for such systems increases. However, due to its multi-disciplinary nature, environmental perception remains an open research topic preventing driverless technology from being widely accepted by the public. A general consensus is that any single sensor is usually unable to provide full knowledge about the environment and a fusion of several modalities is often required. Accordingly, no single reasoning algorithm has been so far capable of handling the complete perception task by itself. When inadvertent sensor failures do occur, perception systems need to adapt to the new operating parameters, that is, the sensor noise model, and continue operating seamlessly. In such cases, it is necessary to detect that a sensor is no longer in its nominal state of work and, whenever possible, identify the true state of work of each sensor in order to avoid dramatic state estimation errors. Finally, intermittent failures to detect an object can also happen even when a sensor is in its nominal state, for example, because of occlusion or ambiguous object configurations. A well-performing perception system should be able to balance the transient changes in operating modes as well as the occasional complete absence of detections.

Due to the vastness of this topic, in this paper, we focus on the difficult problem of detecting and tracking the most vulnerable road users (VRUs) for the purpose of automated collision avoidance and path planning. In the scientific literature, Gavrila et al. [1,2] isolate pedestrians as the most vulnerable road users, arguing that more than 2.5% of the injured pedestrians in collisions with vehicles in Germany, and 4% on the EU level, ended up with fatal consequences. Recent sustainable mobility studies such as the one done by Macioszek et al. [3], illustrating that bike-sharing is on the rise in urban centers, corroborate that safe interaction between automated vehicles and cyclists is becoming equally important. The regulation (EC) 78/2009 [4] defines the key term VRUs as all non-motorized road users such as pedestrians and cyclists as well as motor-cyclists and persons with disabilities or reduced mobility and orientation. Thus, fast and accurate detection and prediction of the position of VRUs is critical in avoiding collisions that often result unfavorably for the VRUs. In that regard, we apply paradigms of modular and real-time design which will later help with the easy integration of the proposed algorithms into a complete perception system.

Tracking-by-detection (TBD) is considered as the preferred paradigm to solve the problem of multiple object tracking (MOT). Algorithms following TBD simplify the complete tracking task by breaking it into two steps: detection of object locations independently in each frame and formation of tracks by connecting corresponding detections across time. Data association, a challenging task on its own, due to missing and spurious detections, occlusions, and target interactions in crowded environments, can be considered to be largely solved with only marginal gains being made in the MOT16 [5] and KITTI [6] benchmarks over the past year. In the tracking literature, it is often assumed that a detection is always available, while in the practice only pragmatic solutions for handling missing detections are being employed. To illustrate this, real-time requirements often force TBD algorithms to discard detections with lower confidence scores which then causes issues in the predict-update cycles of optimal filters. It is therefore quite possible for detection information to be missing in short bursts. When tracking a single target, missing detections can be interpolated using motion and sensor models, however, in multi-object tracking scenarios lost detection information causes ambiguity and is difficult to recover. How such missing detection events should optimally be handled is rarely discussed in the literature, however, the solutions can have a great impact in performance, especially in edge cases.

In this paper we propose a cooperative multi-sensor VRU detection algorithm and tracking algorithm capable of adapting to changing sensors configurations and modes of operation. Moreover, our tracker continues with accurate tracking in cases of missing detections where the lost information is recovered from lower level likelihood information using imputation theory. The proposed algorithms were tuned on a sensor array prototype consisting of a 77 GHz automotive radar and a visible light camera with intersecting fields of view. Each sensor runs its own, high recall, VRU detection neural network providing detection information. The radar detector developed by Dimitrievski et al. [7] outputs dense probability maps in range-azimuth space with peaks at expected VRU positions, while the camera detector of Redmon and Farhadi [8] outputs a rich list of image plane bounding boxes. We use perspective geometry project radar targets on the image plane where they are optimally matched to bounding boxes detected by the camera (Convolutional Neural Network) CNN. Furthermore, we apply sensor to sensor cooperation, details in Section 4.2, using strong radar evidence to improve detection in regions where the camera information is unreliable. Detection to track association is performed by minimizing a matching cost consisting of a distance and appearance likelihood between detections and track maximum aposteriori (MAP) estimates using the Kuhn-Munkres (Hungarian) algorithm [9]. A switching observation model particle filter, details in Section 4.3, handles individual track state estimation by adapting to changes in sensor modes of operation as well as sensor-to-sensor handoff. In cases when detections are missing, the tracker switches to a multiple imputations particle filter and recovers the missing detection from imputations sampled from a novel proposal function conditioned on a target-track likelihood without association, details in Section 4.2. Track maintenance is done based on the evolution of the probabilistic track existence score over time, details in the work of Blackman and Popoli [10].

Experimental evaluation, both in simulation and using data captured in the real-world, shows a significant improvement in detection and tracking performance over intermediate fusion and optimal Bayesian trackers respectively. The proposed cooperative fusion detector is especially effective in low light environments, while the proposed tracker significantly outperforms other tracking algorithms in complex tracking situations with multiple VRUs, heavy occlusion, and large ego-motion. The resultant track estimates remain within tolerable ranges of the ground truth, even in cases where up to 50% of detections are missing. More details behind the ideas inspired in this paper are outlined in the review of relevant literature in Section 2. In Section 3 we provide a condensed overview of the novelties in this paper while in Section 4 we present the detection and tracking algorithms in more detail. We present the experimental methodology, as well as results from simulations and experiments with real sensor data in Section 5. Finally, in Section 6 we give a synopsis of the developed methods and their impact in real-world perception applications.

## 2. Related Work

Multi-sensor multi-object tracking is a wide and interdisciplinary field, so in this section we focus on papers whose ideas inspired the way we implement sensor fusion and handle missing detections during tracking of VRUs. In addition to the summary of each relevant work, we also communicate brief reasoning about how to avoid common mistakes when applying them for applications of autonomous driving perception. Within the application domain of Adaptive Driver Assistance Systems (ADAS), a vast number of non-cooperative sensor fusion techniques exist. Typical examples include the work of Polychronopoulos et al., Blanc et al. and Coué et al., [11,12,13]. An overview of the different approaches to sensor fusion has been already made within the ProFusion2 project in the paper by Park et al. [14]. Of special relevance is their multi-level fusion architecture where raw or pre-processed data coming ‘from different sensors can be found at the signal level. These authors propose the use of back loops between the levels. Back loops can be used to return to a lower level of abstraction for reprocessing certain data or to adapt fusion parameters. This way, processing on adaptive chosen levels can be performed which allows the fusion strategy and the selection of a certain fusion level to be dependent on the actual sensor data and the observation situation of an object. Our camera-radar detector adopts this layered paradigm, where the addition of a feedback loop makes our sensors cooperative, achieving better VRU tracking performance.

In [15] Yenkanchi et al. propose a cooperative ranging-imaging detector based on LiDAR and camera for the task of road obstacle detection. A point cloud is assumed to contain both the ground plane and objects of interest. After a ground plane removal and density-based segmentation step, 3D objects are projected on the camera image plane to form Regions Of Interest (ROIs). The authors use heuristics to fit rectangular masks to various objects such as cars, trucks, pedestrians, and so forth. The resulting image masks are found to visually match with the image content. However it is not clear how these masks can be used for classifying specific targets and whether there is a performance improvement.

Similarly, Gruyer et al. [16] propose a fusion method based on single-line lidar and camera. Detected regions of interest in the lidar data are projected on the camera image plane and instigate tracks. The authors make strong assumptions about the object size in the lidar point clusters which helps to reduce false positives, but only of the objects satisfy the assumptions. Tracking based on belief theory therefore continues by evaluating motion vectors within the projected ROIs. Regions that match with content from past time instances get associated with existing tracks. This approach does not perform object classification of any sort and relies on assumptions about the detected regions based on their size and motion. In addition, it is not a truly cooperative system in a sense that the sensor measurements are not affected by each other.

In the paper [17] by Labayrade et al., a comprehensive cooperative fusion approach using a pair of Stereo Cameras and lidar is presented for the task of robust, accurate and real-time detection of multi-obstacles in the automotive context. The authors take into account the complementary features of both sensors and provide accurate and robust obstacles detection, localization, and characterization. They argue that the final position of the detected obstacles is likely to be the one provided by a laser scanner, which is far more accurate than the stereo-vision. The width and depth will be provided also by laser scanner, whereas the stereo-vision will provide the height of the obstacles, as well as the road lane position. This is a truly cooperative system, since the authors propose a scheme which consists of introducing inter-dependencies: the stereo-vision detection is performed only at the positions corresponding to objects detected by the laser scanner, in order to save computational time. The certainty about the laser track is increased if the stereo-vision detects an obstacle at the corresponding position. In our work we extend this idea to camera-radar cooperative feedback by probabilistic fusion of radar detections to camera objects using real sensor uncertainty models, locally adjusting the detector sensitivity whenever both sensors agree.

The paper [18] by Bao et al. proposes a cooperative radar and infrared sensor fusion technique for the ultimate goal of reduced radar radiation time. They rely on a interacting multiple model (IMM) and an unscented Kalman filter (UKF) to perform tracking whereas the residual of the new information is used to adaptively control the sensor working time. They use the first radar measurement to initialize a track and solve the non-synchronized target detection of the radar and infrared sensors. Furthermore, the probability of switching the Radar on or off is proportional to the residual of the innovation obtained by comparing the filtering result with the estimated measurement. The application of this paper is in aerial target tracking, however, the technique of using the innovation residual for sensor control feedback is directly applicable for automotive systems. Contrary to their approach, due to heavy radar clutter in traffic environments, we instigate tracks based on the camera detection certainty and use the radar to boost this value whenever both sensors see the same target.

In [19], Burlet et al. propose a radar and camera sensor fusion approach for tracking of vehicles. They use a combination of a smart camera and automotive-grade radar in a cooperative fusion scheme. Tracks are initialized from within the narrow Field Of View (FOV) of the radar, but can then be tracked also outside of this FOV as long as they remain visible in the camera image. During an update, each track triggers a raw image search to look for a vehicle in the area where it is predicted to be. Noise from measurements and prediction uncertainties cause the area searched to be bigger than the detected area. The likelihood of the object being a vehicle is calculated using histogram search techniques and evaluating the symmetry of the region. Tests are carried out on highway, rural, and urban scenarios and show a very good detection rate while keeping the number of false positives very low. The paper does not provide details about the cooperative aspect of the image search, which relies on crude edge and symmetry-based object detection. Finally, the paper provides a subjective evaluation of the methods. It does not quantify the effect of fusion on the performance. The general system design is, nonetheless, highly applicable to our perception application.

A late fusion approach was presented by Lee et al. in [20], proposing to match tracking outputs by radar, lidar and camera. The authors propose a Permutation Matrix Track Association (PMTA) which treats the optimal association of tracks from two sensors as an integer programming problem. They relax the association optimization by treating it as a soft alignment instead of hard decision. Thus each entry in the permutation matrix is a value of an objective cost function consisting of spatial, temporal mismatched cost and entropy terms. Of special interest to our approach is the model of the spatial closeness, which these authors design as the joint-likelihood, that is, the product of Radial Basis Functions (RBF) over the distance, velocity and heading of objects. However, the choice of parameters for these functions is not well motivated in the paper.

A maritime object detection and fusion method is proposed by Farahnakian et al. in [21]. The method is based on proposal fusion of multiple sensors such as infrared camera, RGB cameras, radar and LiDAR. Their framework first applies the Selective Search (SS) method on RGB image data to extract possible candidate proposals that likely contain the objects of interest. Then it uses the information from other sensors in order to reduce the number of generated proposals by SS and find more dense proposals. The final set of proposals is organized by considering the overlap between each two data modalities. Each initial proposal by SS is assumed as a final proposal if it is overlapped (Intersection over Union IOU>α) by at least one of the neighboring sensor proposal. Finally, the objects within the final proposals are classified by a Convolutional Neural Network (CNN) into multiple types. In our method, we extend this approach using the Bhattacharyya coefficient a metric since it is theoretically well-founded whereas the IOU has the disadvantage of being non-continuous and zero when two boxes do not overlap.

In [22], Han et al. propose an improvement to the YOLO [23] image object detection algorithm for detection of poorly lit VRUs. They increase the granularity of the YOLO anchor grid in regions where people are detected with low confidence scores. This way, the improved YOLO algorithm can try twice to detect the target at a certain distance according to the characteristic of dim pedestrians and non-motor vehicles. Thus, it can reduce the missing rate of the target and output a more comprehensive scene model and ensure the safe driving of vehicles. This method compensates for a weakness of the original YOLO algorithm where a predefined raster grid is used for region proposal. Our method does not interfere with the way YOLO operates in the sense that we apply feedback from radar only on already detected objects in the image.

Finally, in the past several years there has been an increase of deep-learning-based fusion approaches, especially in the context of autonomous driving. Multiple autonomous car companies have showcased working prototypes based on deep-learning perception, however the specifics of the employed fusion algorithms are unavailable for the public. An exception to this trend is the autonomous research division of Uber which has published several relevant papers by Frossard et al. and Liang et al. [24,25,26]. An organized overview of the currently available scientific literature is presented in the papers by Feng et al. and Wang et al. [27,28] which extensively analyze sensor setups, datasets, fusion methods, and challenges. Of special interest is the analysis of deep fusion networks for multi-modal inputs, where designs ranging from early, middle to late fusion are presented. This overview paper also goes into more detail about techniques based on acyclical network graphs employing various hierarchical shortcuts and feedback loops applied in the works of Chen et al., Ha et al. and Ku et al. [29,30,31]. Most of these techniques rely on re-using a single CNN backbone for computing features while propagating multi-modal information in the ROI estimation CNN blocks (such as the region proposal network or RPN layers). They are, therefore, trained end-to-end on a single dataset which makes it difficult to compare their cross-dataset performance. One typical example is the paper by Mees et al. [32] where they propose an adaptive multi-modal object detection in changing environments. They do so by training a fusion network consisting of an ensemble of detection CNNs, each operating in a specific modality. The fusion net, that is, Mixture of Deep Experts maximizes the probability of classification of a region into the correct category, by weighting the sums of each CNN output. The expert CNN for appearance is pre-trained, while the experts for depth and motion are trained from scratch over the RGB-D People Unihall Dataset [33]. They find that late fusion approach substantially improves detector performance when combining all three modalities by a fully connected layer over a naive way of fusing the individual detectors by averaging the classifier outputs. The performance impact of sensor misalignment as well as cross-dataset performance is not well studied.

Many of the cooperative fusion techniques in the literature perform data fusion by projecting objects detected by lidar/radar onto the image plane. In real-world applications, due to vibrations, the camera and range sensor are not always perfectly calibrated. If the range data has sufficiently dense elevation information, matching of camera and range pixel values can be made robust to such vibrations. When fusing images with radar range information, due to the low elevation resolution of most radars, image and radar detection ambiguities frequently occur. Additionally, when decelerating the car pitches downwards and the radar signal is flooded with clutter from reflections of the ground. The literature rarely discusses this prominent practical problem and its effects remain largely ignored by the likelihood models. Authors have proposed to mitigate these effects by fusing detection information on the ground plane, assuming prior information about the object’s physical dimensions. Specifically, Allebosch et al. [34] present a comprehensive analysis of back-projection techniques for image-to-3D world position estimation was presented. The authors used a single camera to estimate the range of a cyclist based on two different techniques. Firstly, assuming that the wheel of each cyclist is contacting the ground plane, a ray from the camera optical center is cast through the bottom-most pixel of the wheel and the intersection with the physical ground plane is measured. The range found in this manner suffers from wrong estimates of the camera position relative to the world which needs to be re-estimated at run-time. In the second technique, they assume the physical height of the bicycle and use the proportion to the image bounding box size and camera focal length to estimate the distance. In the proposed system, we are detecting and tracking VRUs of unknown height, and therefore relying on prior height assumptions is dangerous. Therefore we perform sensor fusion on the image plane, however in the rare cases where a camera target is not matched to a radar target, we have to rely on range estimation using back-projection where we adopt the techniques discussed in [34].

Sensor-sensor mismatches due to occlusion, sensor failure or various other factors will occur in the real world. When such events do occur, it is of great interest to find an optimal strategy for dealing with missing data. The missing data literature has been so far overwhelmingly focused on regression and classification analysis in big data with missing data in tracking remaining largely understudied. A common practice being the usage of a missing indicator variable and propagation of the past estimate and covariance. However, Pereira et al. [35] demonstrate that in regression and classification problems, methods based on missing-indicator variables are outperformed by ones using imputation theory. This study measures the difference between the missing-indicator method and various imputation methods on 22 datasets when data are missing completely at random. The authors compared classifier performances after applying mean, median, linear regression, and tree-based regression imputation methods with the corresponding performances yielded by applying the missing-indicator approach. The impact was measured with respect to three different classifiers, namely a tree-based ensemble classifier, radial basis function support vector machine classifier, and k-nearest neighbors classifier. In our work, we extend the analysis to the VRU tracking problem where we compare the missing-indicator Kalman and particle filters with a multiple imputations approach.

In the paper [36] by Silbert et al., the authors look at a few track-to-track fusion methods comparing whether it is better to estimate the missing information or ignore it. They use two 2D (single-model) Kalman filter trackers using identical and time-synchronized sensors. Three different target motion behaviors were considered: discrete white noise acceleration, constant velocity, and constant acceleration. Track fusion by three different methods was analyzed: best-of-breed which selects the tracker with minimal covariance at the update, fusion without memory where tracks from individual trackers are combined and the estimate persists only for the current update, and fusion with memory which maintains the combined state estimate and covariance from update to update. This study showed that there is no clear winner meaning that tracking all types of motion depends on the estimation of the process noise and the target motion types. Authors note that accelerating targets are problematic for all methods.

In tracking situations where the observation is polluted with clutter, the probabilistic data association filter (PDAF) has efficiently been studied by Bar-Shalom et al. in [37]. Additionally, the authors also propose a multi-object extension, namely a joint probabilistic data association filter (JPDAF). They argue that the proposed approach is far superior to standard heuristic tracking approaches such as local and global nearest neighbor standard filters (NNSF). They show that in a simulated space-based surveillance example, the PDAF can track a target in a level of clutter in which the NNSF loses the target with high probability. The main approximation for both these algorithms is the single Gaussian used as the state’s conditional probability density function (PDF). Additionally, they argue that the PDAF and JPDAF, using recursive state estimation equations, has far lower complexity than the multiple hypothesis tracker (MHT) in terms of computation time, memory, and code length/debugging. We confirm that in real-time environmental perception systems for autonomous vehicles, tracking based on the MH principle is computationally intractable. Depending on the scene complexity, mainly the number of VRUs being tracked, even the solution of JPDAF is rather complicated since all detection-track pair likelihoods need to be evaluated and updated. To combat this problem, Fansi et al. [38] propose to update the state of multiple tracks by selecting and separately updating groups of targets in interaction. The complexity of the update step is addressed by data association and a gating heuristics. Inspired by both these works, we concluded that no single technique provides desirably high tracking precision as well as low computational complexity. Therefore, we propose a dual approach where a high detection threshold is applied and only confident detections are associated to tracked objects for optimizing execution time. When inadvertent assignment ambiguities or missing detections do occur, we revert to a probabilistic association approach by re-using sub-threshold detection information through imputation theory.

The most relevant analysis on dealing with missing values in non-linear state estimation with particle filters is presented in the work of Housfater et al. and Zhang et al. [39,40]. The authors propose a multiple imputations particle filter formulation that uses randomly drawn values called imputations to provide a replacement for the missing data and then uses the particle filter to estimate non-linear state with the data. Unlike the previous paper, this paper does not assume a linear system and also takes into account the time-varying transition matrix when accounting for missing data. The paper also presents a convergence analysis of the proposed filter and shows that it is almost surely convergent. The performance analysis is based on a simulated non-linear model comparison of the proposed with existing techniques such as extended Kalman filter, sigma point Kalman filter and expectation-maximization algorithm. They conclude that the multiple imputations particle filter has superior performance. This paper inspired our work with the theoretical framework for handling missing data which we extend to a switching observation model and propose a novel proposal function based on the likelihood without association for better conditioning of the sampled imputations on the actual detections.

Missing detections handling by multiple imputations particle filter (MIPF) has been successfully applied in astrometry by Claveria et al. [41]. Albeit defined as a tracking-before-detection, the same principles have been applied in other domains as well. One notable example is underwater acoustic signal processing in the work of Jin et al. [42] where the low signal-to-noise ratio, random missing measurements, multiple interference scenarios, and merging-splitting contacts in measurement space are found to pose challenges for common target tracking algorithms. The authors of this paper propose a tracking-before-detection particle filter that estimates particle likelihood functions directly using the beam-former output energy and adopts crossover and mutation operators to evolve particles with a small weight. The state estimate is therefore largely independent of the availability of detection and significantly outperforms a track-after-detection method based on a Kalman filter. Due to the differences in domains between this paper and our own, the direct application of the proposed method is not possible. However, we adopt the idea of using the raw sensor evidence values as an estimate of the particle likelihood function and draw imputations accordingly when detections are missing.

## 3. System Overview and Novelties

This paper presents a cooperative multi-sensor detection fusion and multi-object tracking system for the estimation of ground plane positions of VRUs. The system is intended to be used as an environmental perception sub-system in autonomous vehicles where making a timely and accurate decision about the presence and position of a person is critical. As such, the design choices which we make optimize both execution speed as well as estimation precision. The system consists of two main components: a cooperative multi-sensor object detector and a multi-object tracker. The detection algorithm, summarized in Algorithm 1, employs intermediate level sensor fusion and sensor-to-sensor cooperative feedback which leads to improved precision and recall in difficult situations, details in Section 4.2. Fusion is performed on the image plane as visualized on Figure 1: first, radar detections, irrespective of their detection scores, are projected on the camera image. Then, we match the most likely camera and radar detections based on the probability that they belong to the same object using the Bhattacharyya coefficient and the appropriate sensor uncertainty models. A fused detection consists of a ground plane position and an image plane bounding box as well as appearance features. The novelty of our detection fusion lies in the sensor-to-sensor cooperative processing where reinforcing weak detection scores in one sensor where there is strong evidence form the other sensor using the sensor uncertainty models. Detections with high detection scores are communicated to the tracker where they are matched to the most likely tracks.
**Algorithm 1** Cooperative camera-radar object detection**Input:** Camera frame $I_{t}$, radar normalized power arrays $P_{t-4:t}$1.  **Detect** radar peaks $\left\{z_{radar,i}\right\}$ from the output of [7]2.  **Detect** camera objects on the ground plane $\left\{z_{cam,i}\right\}$, [8]3.  **Project** radar objects on the image and find matches Equation (Equation 6)4.  **Boost** camera detection scores, Equation (Equation 12)5.  **Back-project** camera objects on the ground plane and fuse with radar Equation (Equation 9)6.  **Fuse** camera and radar detections on the ground plane $z_{fused,i}$ Equation (Equation 9)**Output:** detections:$\left\{\left(z_{fused,i,t},f_{fused,i,t}\right)\right\}$

The proposed tracker, summarized in Algorithm 2, has two modes of operation visualized on Figure 2—initially, the tracker tries to associate a small number of confident detections $z_{fused}$ to existing tracks using a detection-to-track association likelihood based on position and appearance similarity. Depending on the outcome of the association, positively matched detection-track pairs update the state variables using standard Bayesian tracking-by-detection. Detections that cannot achieve the detection threshold or association likelihood criterion are conceptually treated as missing by the tracker. Detections can also be missing due to occlusion, interference, transmission errors or various unknown factors. In cases when detections are expected but not reported, an ill-updated state prediction quickly leads to track divergence. The main novelty of the proposed tracker is an adaptation of the multiple imputations particle filter for handling missing detections, visualized at the bottom of Figure 2. Standard MIPF computes an estimate of the missing detection by sampling imputations from a proposal function defined through the system evolution equations. Since it is difficult to estimate the underlying cause for a missing detection, the standard MIPF proposal function becomes uninformative in the long term. We propose a novel proposal function that is conditioned on the detection likelihood considering all detections without association. This likelihood function is computed at positions indicated by the track estimates using the closest detection from any sensor without applying detection thresholds. Thus, the tracking can satisfy real-time operation requirements as well as an increased robustness to missing detections caused by individual sensor failures or ambiguous data associations. We show that by using the proposed approach the tracker can recover much of the missing detection information by sampling even a single imputation from the proposal. Compared to MIPF, our method achieves greater tracking accuracy while having a reduced computational load, details in Section 4.2. Conceptually, the proposed tracker switches between TBD and JPDAF based on the association outcome which leads to the better utilization of less confident sensor observations.

Due to the uncertain nature of sensor fusion, we treat every fused detection $z_{fused}$ as being generated from a single, fused sensor. However, detections from constituent sensors might not always match, resulting in a detection from a singular sensor. For example, if a camera detection is not matched to any radar detection, the tracker needs to update its state variables using the appropriate camera-only model. We encode the sensor model as a state variable which we also optimize from the accumulated evidence at run-time. We make use of Monte Carlo simulations, where the belief in the state variable is expressed through a set of particles, each switching between different observation models. To the best of our knowledge, the switching observation models particle filter (SOMPF), first proposed by Caron et al. [43], has so far not been implemented for the purpose of multi-sensor VRU tracking. We deem that this method is most suitable for such applications because the switching observation model (SOM) tracker can adapt to misdetections from individual sensors, a common occurrence in automotive perception, details in Section 4.3.
**Algorithm 2** Proposed Multiple Imputations Switching Observation Particle Filter Algorithm1. **Detect objects** (Algorithm 1) **     output:** detections $\left\{\left(z_{j},f_{j}\right)\right\}$, with scores $s_{j}$2. **Track predict:**  **for** each track $\left(x_{t},g_{t}\right)$
**do**   **sample** state particles: ${\tilde{x}}_{t}^{\left(i\right)}∼p_{X|X}\left(x_{t}^{\left(i\right)}|x_{t-1}^{\left(i\right)}\right)$, (Equation 27)   **sample** latent particles:   ${\tilde{\sigma }}_{t}^{\left(i\right)}∼p_{\Sigma |\Sigma }\left(\sigma_{t}^{\left(i\right)}|\sigma_{t-1}^{\left(i\right)}\right);\;{\tilde{a}}_{t}^{\left(i\right)}∼p_{A|A}\left({\tilde{a}}_{t}^{\left(i\right)}|a_{t-1}^{\left(i\right)},{\tilde{\sigma }}_{t}^{\left(i\right)}\right);\;{\tilde{c}}_{t}^{\left(i\right)}∼p_{C|A}\left(c_{t}^{\left(i\right)}|{\tilde{a}}_{t}^{\left(i\right)}\right)$, (Equation 23)–(Equation 25)   **compute** state estimate: ${\tilde{x}}_{t}^{MAP}={\mathrm{argmax}}_{x_{t}}\sum_{i=1}^{N_{pts}}K\left({\tilde{w}}_{t}^{\left(i\right)}{\tilde{x}}_{t}^{\left(i\right)}\right)$3. **Associate:**  **for** each detection $\left(z_{j},f_{j}\right)$
**do**   **for** each track $\left(x_{t},g_{t}\right)$
**do**    **compute** association likelihood: $C_{i,j}=p_{Z|X,C}\left(z_{j},f_{j}|{\tilde{x}}_{i,t}^{MAP},g_{i},c_{t}^{MAP}\right),$  **compute** optimal association list $\Lambda :\left({\left(i,j\right)}_{{\tilde{x}}_{i}^{MAP},z_{j}},...\right)$, [9]4. **Track update**:  **for** each track $\left(x_{t},g_{t}\right)$
**do**   **if**
${\left(i,j\right)}_{{\tilde{x}}_{i}^{MAP},z_{j}}\in \Lambda$
**then**    **update** weights: $w_{t}^{\left(i\right)}\leftarrow w_{t-1}^{\left(i\right)}p_{Z|X,C}\left(z_{j},f_{j}\left|{\tilde{x}}_{t}^{\left(i\right)},g_{i},{\tilde{c}}_{t}^{\left(i\right)}\right.\right),$ (Equation 47)    **update** existence score: $\chi_{t}\leftarrow \chi_{t-1}-\ln\left[1+\exp\left(-2\Lambda_{\left(i,j\right)}\right)\right]-\ln\left[p_{null}\left(x_{0:t}\right)\right]$, [10,44]   **else**
*%missing observation*    **sample** an imputed detection: $y_{t}^{\left(j\right)}∼\phi_{Y|Z}\left(y_{t}\left|,z_{t}\right.\right)\;c_{t}^{\left(j\right)}\leftarrow 0$,    **update** weights: $w_{t}^{\left(i\right)}\leftarrow \sum_{j=1}w_{t}^{\left(i,j\right)}=w_{t-1}^{\left(i\right)}L_{fused,t}\left(y_{t}^{\left(j\right)}\right)$    **update** existence score: $\chi_{t}\leftarrow \chi_{t-1}-\ln\left[Pr\left(c_{t}=0\right)\right]$, [10,44]5. **Re-sample:**  **for** each track $\left(x_{t},g_{t}\right)$
**do**   **normalize** weights: $w_{t}\leftarrow w_{t}/{∥w_{t}∥}_{2}$   **if** effective sample size: $N_{eff=}1/\sum_{i}{\left(w_{t}^{\left(i\right)}\right)}^{2}<0.2$
**then**    **sample**
$i\in \left\{1,...,N_{pts}\right\}$ with probability $\propto w_{t}^{\left(i\right)}$ and replace $x_{t}^{\left(i\right)},a_{t}^{\left(i\right)},c_{t}^{\left(i\right)}$6. **Spawn tracks:**  **for** each $\left(z_{j},f_{j}\right);j\notin \Lambda$
**do***%unassociated detection*   **if**
$s_{j}\geq \tau_{spawn}$
**then**    **initialize** a new track $\left(x_{t},g_{t},a_{t},c_{t},w_{t},\chi \right)≜\left\{\left\{x_{t}^{\left(i\right)}\right\},\left\{c_{t}^{\left(i\right)}\right\},\left\{a_{t}^{\left(i\right)}\right\},\left\{w_{t}^{\left(i\right)}\right\},g_{t}\right\}:$     **sample** particles $x_{t}^{\left(i\right)}∼N_{2}\left(\mu_{z_{j}},\sum_{0}\right);$     **sample** particles $a_{t}^{\left(i\right)}∼p_{A|\Sigma }\left(a_{0}|\sigma_{0}\right)$ and compute $c_{t}^{\left(i\right)}$     **set** weights $w_{t}^{\left(i\right)}\leftarrow 1/N$     **set** existence score: $\chi_{t}=-\ln\left[1+\exp\left(-2s_{j}\right)\right]-\ln\left[p_{null}\left(x_{0}\right)\right]$, [10,44]7. **Remove tracks:**  **for** each track $\left(x_{t},g_{t}\right)$
**do**   **if $\chi_{t}<\tau_{\chi_{0}}$ then**    $\left(x_{t},g_{t},a_{t},c_{t},w_{t},\chi \right)≜\emptyset$

## 4. Proposed Method

### 4.1. Concepts

The perception system considered in this paper consists of imperfect sensors that produce noisy measurements and give a distorted picture of the environment. The goal of the overall system is to estimate the position and velocity of a generally unknown number of vulnerable road users by temporal integration of noisy detections. We model VRUs as the unordered set of tuples $V\left(t\right)≜\left\{\left(x_{i},g_{i}\right)\right\}$ consisting of a state vector $x:\left(\rho ,\theta \right)$ containing the VRU position in polar coordinates and a feature vector g containing various information unique to the person such as their velocity vector, visual appearance, radar cross section and so forth. At the same time, each sensor *k* generates a set of measurements $D_{k}\left(t\right)≜\left\{\left(z_{k,j},f_{k,j}\right):k=0...K\right\}$ which is an unordered set of tuples $\left(z_{k,j},f_{k,j}\right),$ where $z_{k,j}$ is a vector indicating the location of a detection in sensor coordinates and the vector $f_{k,j}$ contains the reliability of the detection and all features other than its position, for example, the width and height of its bounding box, an estimated Doppler velocity and radar signal strength, an estimated overall color, and so forth. The number of elements in all of these vectors is always the same for a given sensor (it does not vary in time), but can differ from sensor to sensor (e.g., a radar can output different and more or fewer features than a camera). After sensor fusion, we obtain a set of confident detections $D_{fused}\left(t\right)≜\left\{\left(z_{fused,j},f_{fused,j}\right)\right\}$ and the goal of the tracker is to estimate the state vectors $x_{i,t}\in \left\{x_{1,t},...,x_{n,t}\right\}$ at time *t*, integrating current and previous confident detections $z_{j,1..t}$ by means of optimal matching between detections and VRU tracks. Association of detections $D\left(t\right)$ to tracks $V\left(t\right)$ is performed by searching for the globally optimum association solution using a detection-to-track likelihood $p_{Z|X}\left(z_{j},f_{j}|x_{i},g_{i}\right)$ as a metric. This likelihood is a product of the ground plane positional likelihood $p_{Z|X}\left(z_{j}|x_{i}\right)$ and the image plane likelihood of feature vectors $f_{j}$ and $g_{i}$. Whenever a track $\left(x_{i},g_{i}\right)$ is not associated with a detection, then the state variable $x_{i}$ is updated using an imputed detection $y_{j}\approx z_{j}$. In our proposed method, we sample $y_{j}$ using the likelihood without association $p_{Z|X}\left(z_{j}|\left(\rho_{i},\theta_{i}\right)\right)\approx \phi_{Y|Z}\left(y_{j}|\left(\rho_{i},\theta_{i}\right)\right)=L_{fused};\forall \left(\rho_{i},\theta_{i}\right),\forall z_{j}$ which we precompute during detection. This sampling has the effect that, in ambiguous cases, multiple individual detections can update multiple unassociated tracks.

### 4.2. Cooperative Sensor Fusion

We base our analysis on fusion of noisy camera and radar detections, but the method can easily be extended to any sensor configuration. For a camera, a detection $z_{cam}$ contains the center coordinate of a bounding box in the image, while for radar $z_{radar}$ contains the polar coordinates of a radar detection in the radar coordinate system. The diagram on Figure 3 presents one typical sensor layout designed for environmental perception in front of a vehicle. Due to the different sensor fields of view (FoV), different regions of the environment are covered by none, one or both sensors. Moreover, it can be expected that the characteristics of sensor operation change locally depending on the scene layout. For the camera, every occluding object creates blind spots where misdetection is likely. On the example layout on Figure 3, the person in red cannot be detected by the camera because of occlusion. However, this occluded region in the camera, although attenuated, is still visible by radar. Using radio wave propagation principles we can expect that any occluding object within the first Fresnel zone, an ellipsoid with radius:(1)$$F_{1}=\frac{\sqrt{\frac{cD}{f}}}{2},$$
will degrade the signal strength. $F_{1}$ is defined through *c*, the speed of light in the medium, the distance *D* between the transmitter and receiver, and *f* the wave frequency. In practice, the first Fresnel zone should be at least 60% free of obstacles for a good signal. For a person standing at 20 m in front of the radar, the radius of the first Fresnel zone for a 77 GHz radar beam is roughly 20 cm. If a person is occluding this first Fresnel zone, it will not create a complete radar occlusion because gaps between the parts of the body allow for signal propagation. The radar signal will nevertheless be attenuated. Additional problems for radar detection come from the effect of multipath propagation caused by reflections from flat surfaces (walls, large vehicles, etc.). On the diagram on Figure 3 this is visualized as a hole in the radar frustum near the flat surface of a truck. In such areas, the radar signal fades significantly and detection rates reduce accordingly. Very near to flat surfaces, the radar mode of operation can even switch to a degenerate one even though the area is well within the radar FoV. Incorporating such prior information about sensor coverage zones requires accurate knowledge of the 3D scene. This knowledge is difficult to compute in real-time systems as it depends upon ray-tracing of camera as well as radar signals in the scene. Instead of precisely modeling the frustums of each sensor, we let the SOMPF tracker determine the sensor operating mode from the characteristics of the associated detections over time. To that end we propose an intermediate-level fusion technique for integrating radar and camera information prior to applying detection threshold, which results in a set of fused measurements $D_{fused}\left(t\right)≜\left\{\left(z_{fused,j},f_{fused,j}\right)\right\}$ as if they were produced from a single sensor with multiple, locally varying, observation models. This fused sensor covers the area of the union of original sensors. Thus, the problems which arise in modeling intersections and unions of sensor frustums are handled elegantly in a single model.

Detection starts by running independent object detection CNNs on the raw image and radar data. The employed camera detector, YOLOv3 [8], outputs a set of bounding boxes, while the radar detection CNN [7] outputs an array of VRU detection scores in polar coordinates, visualized on Figure 1. We interface with both detectors prior to applying a detection threshold and obtain a complete set of detection candidates. These candidate detections can be considered as intermediate-level outputs as they exert a high recall rate, but also contain many false positives. Due to their high recall rate, objects detected confidently by one sensor, but not so well in the other can be used to reinforce one another’s detection scores. This concept of sensor-to-sensor cooperation can help improve detection rates in areas of poor visibility for one sensor and will be explained in more detail in Section 4.2. After applying the sensor to sensor cooperative logic, we fuse the camera and radar mid-level detections by projecting and matching on the image plane. Detection fusion yields fused detections $z_{fused,j}$ which are matched based on proximity on the image, while their respective feature vectors $f_{fused,j}$ are formed using the feature vectors of the closest camera and radar target. We note that due to calibration and detection uncertainty, a fused detection can often contain only camera $\left(\left(z_{fused},f_{fused}\right)=\left(z_{cam},f_{cam}\right)\right)$ or radar $\left(\left(z_{fused},f_{fused}\right)=\left(z_{radar},f_{radar}\right)\right)$ evidence depending on how well the two sensors agree.

Compared to early fusion, our intermediate fusion algorithm reduces the data transfer rate, but retains most of the information useful for object tracking. Both object detection CNNs are built on the same hierarchical principle, that is, they first produce mid-level information consisting of object candidates (proposals) which are then further classified into VRUs. We detect radar detection candidates $z_{radar,i}$ within small neighborhoods $S_{i}$ by means of non-maximum suppression of the radar CNN output, formally where every local maxima $z_{radar,i}\in \left\{z_{radar,1},...,z_{radar,n_{radar}}\right\}$ is a vector consisting of the range and azimuth $z_{radar,i}={\left(\rho_{radar,i},\theta_{radar,i}\right)}^{T}$ of the detection peak and has a confidence score $f_{radar,i}=s_{radar,i}$ equal to the peak’s height. The radar detections are distributed according the following observation model:(2)$$z_{radar,t}=x_{t}+w_{radar,t};\;w_{radar}∼N_{2}\left(0,R_{1}\right),$$
where x is the VRU position on the ground plane expressed in polar coordinates, $x={\left(\rho ,\theta \right)}^{T}$ and $N_{2}\left(0,\cdot \right)$ is a zero-mean bi-variate Gaussian distribution with a covariance matrix $R_{1}$ of known values $\mathrm{diag}\left(\sigma_{radar,\rho }^{2},\sigma_{radar,\theta }^{2}\right)$. Using the extrinsic and intrinsic calibration matrices K,R,T we project each radar detection on the image plane where a detection maps to pixel coordinates $\left(u_{radar,i},v_{radar,i}\right)$. Naturally, the uncertainty in azimuth projects to horizontal uncertainty on the image plane, thus radar detections have a much higher positional uncertainty in the horizontal direction than in the vertical. This uncertainty can be computed by propagating the ground plane covariance matrix $R_{1}$ to image coordinates and estimating the image covariance matrix $R_{1,IP}$ of the newly formed random variables $\left(u_{radar,i},v_{radar,i}\right)$. Due to the non-linearity of this transform, an interval propagation can be performed in order to compute intervals that contain consistent values for the variables using the first-order Taylor series expansion. The propagation of error in the transformed (non-linear) space can be computed using the partial derivatives of the transformation function. However, we found out that the parameters of the covariance matrix of radar projections on the image plane, $R_{1,IP}$, can accurately be approximated by the following transformation:(3)$$R_{1,IP}=\mathrm{diag}\left(\sigma_{radar,u}^{2},\sigma_{radar,v}^{2}\right)=\mathrm{diag}\left(2f_{y}\tan\left(\frac{\sigma_{radar,\theta }^{2}}{2}\right),\frac{f_{x}\sigma_{radar,\rho }^{2}}{\rho_{radar}}\right),$$
where $f_{x}$ and $f_{y}$ are the horizontal and vertical camera focal lengths and ρ is the range of the radar target. Thus, for radar targets $z_{radar}^{IP}$ on the image plane we can use the following observation model:(4)$$z_{radar,t}^{IP}=x_{t}^{IP}+w_{radar,t}^{IP};\;w_{radar}^{IP}∼N_{2}\left(0,R_{1,IP}\right),$$
where $x_{t}^{IP}$ is the vector of image plane coordinates of the feet of a person at time *t* and $R_{1,IP}$ is the radar covariance matrix in image coordinates $R_{1,IP}=\mathrm{diag}\left(\sigma_{radar,u}^{2},\sigma_{radar,v}^{2}\right)$.

Similarly, we interface with the image detection CNN at an early stage where we gain access to all detections regardless of their detection score. Each camera detection consists of the image coordinates of the bounding box center as well as the BB width, height, appearance vector and a detection score: $f_{cam,i}=\left(width,height,app,s_{radar,i}\right)$. For modeling the image plane coordinates of the feet of a person detected by a camera detector, $z_{cam}^{IP}$, we use the following model:(5)$$z_{cam,t}^{IP}=x_{t}^{IP}+w_{cam,t}^{IP};\;w_{cam}^{IP}∼N_{2}\left(0,R_{2,IP}\right),$$
where the covariance matrix $R_{2,IP}=\mathrm{diag}\left(\sigma_{cam,u},\sigma_{cam,v}\right)$ can be estimated offline from labeled data.

In order to match camera detections to radar detections on the image plane, we rely on the Bhattacharyya distance as a measure of the similarity of two probability distributions. Practically, we compute the Bhattacharyya coefficient of each camera and radar detection pair, which is a measure of the amount of overlap between two statistical samples. Using the covariance matrices $R_{1,IP}$ and $R_{2,IP}$ of the two sensors, the distance of any two observations $\left(z_{cam}^{IP},z_{radar}^{IP}\right)$ is computed as:(6)$$BC\left(z_{cam},z_{radar}\right)=\exp\left(-d_{B}\left(z_{cam}^{IP},z_{radar}^{IP}\right)\right),$$
where $d_{B}$ is the Bhattacharyya distance:(7)$$d_{B}\left(z_{cam}^{IP},z_{radar}^{IP}\right)=\frac{1}{8}{\left(z_{cam}^{IP}-z_{radar}^{IP}\right)}^{T}R_{\mathrm{IP}}^{-1}\left(z_{cam}^{IP}-z_{radar}^{IP}\right)+\frac{1}{2}\ln\left(\frac{\det R_{\mathrm{IP}}}{\sqrt{\det R_{1,IP}\det R_{2,IP}}}\right),$$
and the covariance matrix $R_{IP}$ is computed as the mean of the individual covariance matrices: $R_{IP}=\frac{R_{1,IP}+R_{2,IP}}{2}.$ Closely matched detections form a fused detection $\left(z_{fused,i},f_{fused,i}\right)$ whose image features $f_{fused}$ (BB size and appearance) are inherited from the camera detection. For computing the ground plane position $z_{fused}$, we need to project both camera and radar detections to the ground plane. Due to the loss of depth in the camera image, a camera detection $z_{cam,i}$ has ambiguous ground plane position. We can, however, use an estimate based on prior knowledge. Specifically, we make the assumption that the world is locally flat and the bottom of the bounding box intersects the ground plane. Then, using the intrinsic camera matrix we back-project this intersection from image to ground plane coordinates. This procedure achieves satisfactory results when the world is locally flat and the orientation of the camera to the world is known. In a more general case, back-projection will result in significant range error which we model with the following observation model:(8)$$z_{cam,t}=x_{t}+w_{cam,t};\;w_{cam}∼N_{2}\left(0,R_{2}\right),$$
where the the covariance matrix $R_{2}=\mathrm{diag}\left(\sigma_{cam,\rho }^{2},\sigma_{cam,\theta }^{2}\right)$ consists of a radial standard deviation part $\sigma_{cam,\rho }$, a function of the distance:$$\sigma_{cam,\rho }=ax\left(t\right)+b.$$
and an azimuth standard deviation parameter $\sigma_{cam,\theta }$. Having both the radar and camera detection positions on the ground plane, we combine them using the fusion sensor model explained by Willner et al. and Durrant-Whyte et al. [45,46]:(9)$$z_{fused,i}=R_{3}\left(R_{1}^{-1}z_{radar,i}+R_{2}^{-1}z_{cam,i}\right),$$
where the covariance matrix $R_{3}$ is computed as:(10)$$R_{3}^{-1}=\sum_{i=1}^{2}R_{i}^{-1}.$$

In practice, depending on how closely matched the two constituent detections are, a fused detection will obtain the range information mostly from the radar and the azimuth information mostly from the camera. On the other hand. the further apart the two constituent detections are, the more the fused target will behave like a camera-only or a radar-only target. In the special case when a camera detection is not matched to a radar detection (BC∼0), the resulting fused detection will consists of only the assumed camera range and azimuth and the camera detection features $\left(\left(z_{fused},f_{fused}\right)=\left(z_{cam},f_{cam}\right)\right)$. Similarly, an unmatched radar detection yields a fused detection consisting of the radar range and azimuth, radar detection score and an assumed image plane BB $\left(z_{fused},f_{fused}\right)=\left(z_{radar},f_{radar}\right)$. In these cases, a fused detection is explained purely by the individual sensor model of either the camera Equation (Equation 8) or the radar Equation (Equation 2). Following the fusion method explained in [45] the detection score is averaged from the camera and radar detection scores: (11)$$s_{fused}=\frac{s_{cam}+s_{radar}}{2}.$$

A potential weakness of averaging the detection scores of individual sensors becomes obvious when the operating characteristics of one sensor become less than ideal. We therefore propose a smart detection confidence fusion algorithm with the key idea of using the strengths of one sensor to reduce the other one’s weakness. The proposed algorithm produces significantly better detection scores in regions of the image with poor visibility caused by either low light, glare or imperfections and deformations on the camera lens. This is because radar can effectively detect any moving object regardless of the light level and its detection score be used to reinforce the weak image detection score. For closely matched pairs of detections where one sensor’s detection score is below a threshold, we will increase the detection score proportionally to the proximity and confidence of the other sensor’s detection. The resulting candidate object list recalls a larger amount of true positives at comparable false alarm rate.

Specifically, objects with a low detection score $s_{cam}<\tau_{det}$, which are also detected by Radar $BC\left(z_{cam},z_{radar}\right)∼1$, will have an increased detection confidence proportional to the similarity coefficient in Equation (Equation 6), formally: (12)$$s_{cam}\max\left(s_{cam},\sqrt{\tau \left[s_{cam}+\left(1-s_{cam}\right)BC{\left(z_{cam},z_{radar}\right)}^{\beta }\right]}\right),,$$
where the parameter β controls the magnitude of reinforcement. Finally, after fusion and boosting, confident detections $\left(z_{fused,i},f_{fused,i}\right);\forall s_{fused,i}>\tau_{det}$ are fed to the tracker where they are associated to a track $\left(x_{i},g_{i}\right)$ by optimizing the global detection to track association solution based on the association likelihood $p_{Z|X}\left(z_{j},f_{j}|x_{i},g_{i}\right)$.

On Figure 4 we present typical examples where the proposed cooperative fusion method brings significant improvements. In this poorly lit garage, people can be frequently seen exiting on foot from parked cars or on a bicycle from a bicycle parking area. Even in very low ambient light, the radar is able to accurately detect motion and thus boost low detection scores in suspected regions detected by the camera. Further numerical analysis of the performance is given in Section 5 where we compare against a control algorithm that does not apply confidence-boosting.

### 4.3. Tracking by Detection Using Switching Observation Models

Assuming that at time *t* the multiple detections $z_{fused,j}$ are optimally associated to tracks $x_{i}$ by a detection-track association mechanism we will hereby explain the single target tracking algorithm which maximizes the belief of the state of a VRU by temporal integration of noisy measurements from multiple sensors. For the sake of notational simplicity we will drop the index “fused” assuming that all detections underwent the data fusion steps in Algorithm 1. The state vector $x_{t}$ consisting of the persons position and on the ground can be estimated from the probability density function (PDF) $p_{X|Z}\left(x_{0:t},\left|z_{1:t}\right.\right),$ by aggregating detections $z_{1:t},$ over time. Due to the real-time nature of autonomous vehicle perception, we assume the Markov property whenever possible and confine to using recursive, single time-step operations. The process is governed by a state evolution model:(13)$$x_{t}=f\left(x_{t},v_{t}\right),$$
consisting of the nonlinear function *f* and noise term $v_{t}$ with known probability density $p_{V}\left(v\right)$. The true nature of *f* which explains the motion of a person on the ground plane, is dependent on both scene geometry as well as high-level reasoning which is difficult to model. Oftentimes in high frame-rate applications, human motion is commonly modeled using a constant velocity motion model which accurately explains human motion over such short time intervals. Each detection $\left(z_{t},f_{t}\right)$, is related to a track $\left(x_{t},g_{t}\right)$ through the observation probability, or likelihood laws $p_{Z|X}\left(z_{t,j},f_{t,j}|x_{t,i},g_{t,i}\right)$ discussed in Equations (Equation 2), (Equation 8) and (Equation 9). The spatial component $p_{Z|X}\left(z_{j}|x_{i}\right)$ models the likelihood that that an observation $z_{j}$ stems from the state vector $x_{i}$ on the ground plane, while the image component $L_{IP}\left(f_{j}|g_{i}\right)$ models the likelihood of observing the image features $f_{j}$ for the specific track $\left(x_{i},g_{i}\right)$ with features $g_{i}$, formally:(14)$$p_{Z|X}\left(z_{j},f_{j}|x_{i},g_{i}\right)=p_{Z|X}\left(z_{j}|x_{i}\right)L_{IP}\left(f_{j}|g_{i}\right),$$
where $p_{Z|X}\left(z_{j}|x_{i}\right)$ is usually inversely proportional to the Mahalanobis distance of the detection $z_{j}$ to the sensor model centered around the posterior estimate $x_{i}^{\mathrm{MAP}}$:(15)$$p_{Z|X}\left(z_{j}|x_{i}^{\mathrm{MAP}}\right)=\exp\left(-\frac{1}{2}{\left(z_{j}-x_{i}^{\mathrm{MAP}}\right)}^{T}R_{3}{\left(z_{j}-x_{i}^{\mathrm{MAP}}\right)}^{T}\right),$$
and $L_{IP}\left(f_{j}|g_{i}\right)$ is a function measuring distance of image features such as BB overlap, shape or color similarity. In our previous work [47] we have shown that an image plane likelihood consisting of the Jaccard index and the Kullback-Leibler divergence of HSV histograms between two image patches can be used as an excellent detection to track association metric, thus formally:(16)$$L_{IP}\left(f_{j}|g_{i}\right)=IOU\left(f_{j}|g_{i}\right)KL\left(f_{j}|g_{i}\right),$$
where IOU measures the intersection over union of the detection and track bounding boxes and while KL is the Kullback-Leibler divergence of the detection and track HSV color histograms.

Depending on the external or internal conditions, our fused sensor can be operating in a specific state *c* described by its specific observation function. We model the sensor state using the latent variable *c* such that it can switch between categorical values explaining different sensing modalities. For example, detection of objects in good conditions can be considered to be a nominal state c=1 and the detections can be explained through the nominal likelihood $p_{Z|X,C}\left(z_{t},f_{t}\left|x_{t},g_{t},c_{t}\right.\right)$ in Equation (Equation 14). In the further analysis we will drop the detection and track feature vectors, f and $g_{}$, for the sake of notational simplicity. During operation, conditions can change dramatically due to changing light levels, transmission channel errors, battery power level. A camera will react to such changes by adjusting it’s integration time, aperture, sensitivity, white balance and so forth, which inadvertently results in detection characteristics different from the nominal ones. Any sensor, in general, can stop working altogether in cases of mechanical failure caused by vibrations, overheating, dust contamination and so forth. Additionally, manufacturing defects, end of life cycle, physical or cyber-attacks can alter the characteristics of measured data. Lastly, real-world sensor configurations employed to measure a wide area of interest often have “blind spots” where information is missing by design. All these factors can influence the current observation likelihood to be different from a nominal one.

For optimal tracking it is important to have a good model of the various modes of operation the sensor. Ideally, the chosen observation model should have the parameter flexibility to be able to adapt itself according to the gradual change of operation. Due to practical limitations in real-world operation, we are inclined to model only the most relevant sensor modes using discrete values for *c*, for example a day-time and night-time camera characteristics or uncluttered and cluttered radar environment characteristics. We therefore use a list of observation models $p_{Z|X,C}\left(z_{t}\left|x_{t},c_{t}\right.\right);c=1,...,n_{c},$ where the appropriate sensor model *c* is chosen at run-time through optimization. Thus, the sensor may switch between states of operation $c\in \left\{0,...,n_{c}\right\}$. In a general case, for each sensor k≥1 there are $n_{k,c}$ possible observation models, however since we are using detection fusion, our fused sensor can switch between the modes of operation of every constituent sensor:(17)$$z_{t}=h_{c_{t}}\left(x_{t}\right)+w_{t,c_{t}},$$
where $h_{c_{t}}$ is a nonlinear observation function and $w_{t,c_{t}}$ is an observation noise of the observation model $c_{t}$ at time *t*. The following modes of operation are possible:(18)$$c_{t}=\left\{\begin{array}{cc} 0 & \mathrm{if}\mathrm{the}\mathrm{fused}\mathrm{detection}z_{t}\mathrm{is}\mathrm{independent}\mathrm{of}x_{t}\\ 1 & \mathrm{if}\mathrm{the}\mathrm{fused}\mathrm{sensor}\mathrm{is}\mathrm{in}\mathrm{its}\mathrm{nominal}\mathrm{state}\mathrm{of}\mathrm{work}\\ j,\;j\in \left\{2,...,n_{c}\right\} & \mathrm{if}\mathrm{the}\mathrm{fused}\mathrm{sensor}\mathrm{is}\mathrm{in}\mathrm{its}j-\mathrm{th}\mathrm{state}\mathrm{of}\mathrm{work} \end{array}\right.$$
meaning that when a sensor is in a failure mode, $z_{t}$ is statistically independent of $x_{t}$, the degenerate model $p_{Z|X,0}\left(\cdot |\cdot \right)$ is used. In a nominal state of work $c_{t}=1$ a sensor is assumed to be producing detections as it was indented by design, while in the other *j* states of work the sensor is producing various levels of service. It is important to note that the sensor mode of operation also varies across the field of view. For camera and radar CNN object detectors this means that the detection quality will change in different regions of the field of view due to transient occlusions, light changes due to shadows or multi-path reflections which can cause an object detection score to briefly drop below the detection threshold. To complete the system we use a set of confidences that our fused sensor is in a certain state $\left\{a_{j}\right\}$ which model the probability that the sensor is in a given operational state $j,\;j\in \left\{0,...,n_{c}\right\}$:(19)$$a_{j,t}=Pr\left(c_{t}=j\right),\;a_{j,t}\geq 0,\;\sum_{j=0}^{n_{c}}a_{j,t}=1.$$

Authors in [43] state that these confidences are difficult to know a priori due to the possibility of rapid changes of external conditions and propose to tune them adaptively using a Markov evolution model. Thus the transition of probabilities $a_{j}$ over time is:(20)$$a_{t}∼p_{A|A}\left(a_{t}\left|a_{t-1}\right.\right),$$
with $a_{t}$ being the vector consisting all individual $a_{j}$ at time *t*. The posterior PDF over the time interval $\left[0,t\right]$ given the switching observation model formulation expands to:(21)$$p_{X,A|Z}\left(x_{0:t},a_{0:t}\left|z_{1:t}\right.\right),$$
where of practical interest is the marginal posterior at current time $p_{X,C|Z}\left(x_{t},c_{t}\left|z_{1:t}\right.\right)$, while the maximum a posteriori estimates $x_{t}^{MAP}$ and $c_{t}$ can be communicated to the chain of perception systems as the best estimates of a VRU position and the state of sensor at time *t*. For estimation of the state vector $x_{t}$ we rely on the estimation of the latent variable $c_{t}$ for which, assuming they both have Markov property, Bayesian tracking can be employed. We adopt the same evolution model structure as proposed by [43]: (22)$$\begin{array}{c} x_{t}∼p_{X|X}\left(x_{t}|x_{t-1},\sigma_{t}\right),\\ c_{t}∼Pr\left(c_{t}|a_{t}\right),\\ a_{t}∼p_{A|A}\left(a_{t}|a_{t-1},\sigma_{t}\right),\\ \sigma_{t}∼p_{\Sigma |\Sigma }\left(\sigma_{t}|\sigma_{t-1}\right), \end{array}$$
where the Markov property of $a_{t}$ results in a sensor state $c_{t}$ dependent on its past values marginalized over $a_{0:t}$. The prior over the sensor state variables $c_{t}$ being:(23)$$Pr\left(c_{t}|a_{t}\right)=\sum_{j=0}^{n_{c}}\delta_{j}\left(c_{t}\right)a_{j,t}$$
where the vector $a=\left\{a_{0},...,a_{n_{c}}\right\}$ needs to be estimated as well since it can vary over time and over the field of view of the sensors, for example, the reliability of a sensor decreasing over time. Because $a_{t}$ models the confidence of the state of the sensor, and a sensor can only be in one state at a time, Equation (Equation 19), we can effectively use the $n_{c}$-dimensional Dirichlet distribution $D\left(\alpha_{1},...,\alpha_{n_{c}}\right)$ as a model for the evolution $p_{A|A}\left(a_{t}|a_{t-1},\sigma_{t}\right)$. The intuition behind this distribution lies in the interpretation of the concentration vector α as a measure of how concentrated the probability of a sample will be. For example, if $\alpha_{i}<1$ the sample is very likely to fall in the *i*-th component, that is, the sensor to be in that mode of operation. If $\alpha_{i}>1$ then the uncertainty of the sensor state will be dispersed among all components. For our fused sensor we propagate the confidences $a_{t}$ from $a_{t-1}$ as:(24)$$p_{A|A}\left(a_{t}|a_{t-1},\sigma_{t}\right)=D\left(\sigma_{t}^{a}a_{0,t-1},...,\sigma_{t}^{a}a_{n_{c},t-1}\right),$$
using the spread parameter $\sigma_{t}^{a}$, that adjusts the spread of probabilities $a_{j,t}$, which is estimated using the evolution model:(25)$$\log\left(\sigma_{t}^{a}\right)=\log\left(\sigma_{t-1}^{a}\right)+\lambda^{a},$$
where $\lambda^{a}$ is a zero-mean white Gaussian noise with known variance. The hyper-parameter transition function follows the density $p_{\Sigma |\Sigma }\left(\sigma_{t}|\sigma_{t-1}\right)$ and the authors in [43] propose to use a Gaussian noise model with variances $\sigma_{t}^{v}$ that are also estimated. To reduce the complexity of the estimation process, in our approach we use fixed variances while the log function in Equation (Equation 25) is used to ensure that the variances remain positive.

Finally, for the state evolution function *f* in Equation (Equation 13) we use a short-time behavioral motion model learned from annotated pedestrians walking in an urban environment. The current position and velocity is propagated from the past state using random changes in the longitudinal and lateral velocities, expressed in polar coordinates:(26)$$v_{\rho ,t}∼N\left(v_{\rho ,t-1},\sigma_{\rho }\right),\;v_{\theta ,t}∼N\left(v_{\theta ,t-1},f\left(v_{\rho ,t}\right)\right),$$
(27)$$\rho_{t}=\rho_{t-1}+\Delta v_{\rho ,t},\;\theta_{t}=\theta_{t-1}+\Delta v_{\theta ,t}.$$

We refer the reader to find more details about this motion model in [47].

### 4.4. Sampling-Based Bayesian Estimation

In applications such as VRU tracking in traffic scenarios, the posterior in Equation (Equation 21) has a highly complex shape (often multi-modal) and cannot be computed in a closed form. For example, when a person becomes occluded for an extended time period, it is desirable to allow for multiple hypotheses to exist so that the same person can be accurately re-identified when detection evidence arrives. Additionally, camera and radar observation models are highly non-linear, making classical linear filters such as Kalman to become ineffective [48]. To that end, we use the sequential Monte Carlo (SMC) method called particle filter (PF) which provides a numerical approximation of the state vector PDF using a set of weighted samples (particles). The tradeoff between estimation accuracy and computational load can be tuned by adjusting the number of particles in the filter. In a standard particle filter, each particle ${\tilde{x}}_{0:t}^{\left(i\right)}$ with its corresponding weight ${\tilde{w}}_{t}^{\left(i\right)}$ approximates the posterior PDF $p_{X|Z}\left(x_{0:t}\left|z_{1:t}\right.\right)$ through the empirical distribution $P_{N}\left(dx_{0:t}\right)$ as:(28)$$P_{N}\left(dx_{0:t}\right)=\sum_{i=1}^{N_{pts}}{\tilde{w}}_{t}^{\left(i\right)}\delta_{{\tilde{x}}_{0:t}^{\left(i\right)}}\left(dx_{0:t}\right),$$
where $\delta_{x_{0:t}^{\left(i\right)}}\left(dx_{0:t}\right)$ is the delta–Dirac mass located in $x{\left(i\right)}_{0:t}$. This distribution can be used to compute an estimate of the state vector, for example, the minimum mean squared error (MMSE) estimate is given as:(29)$$E_{p_{X|Z}\left(x_{0:t}\left|z_{1:t}\right.\right)}\left[x_{0:t}\right]=\int_{X^{t+1}}x_{0:t}p_{X|Z}\left(x_{0:t}\left|z_{1:t}\right.\right)dx_{0:t}\approx \int_{X^{t+1}}x_{0:t}P_{N}\left(dx_{0:t}\right)=\sum_{i=1}^{N_{pts}}{\tilde{w}}_{t}^{\left(i\right)}{\tilde{x}}_{0:t}^{\left(i\right)}.$$

In practice however, it is more desirable to compute the maximum a posteriori (MAP) estimate of the PDF. It is easy to imagine a situation when the posterior PDF becomes strongly multi-modal, for example, when a person becomes occluded and can follow one of several possible paths of motion. The expected state PDF of such a person is then multi-modal with high probabilities for each model mean. A MMSE estimate will give a wrong result, so it becomes more useful to compute the MAP estimate $x_{0:t}^{MAP}$ through one of the many mode finding techniques. For example, kernel density estimation (KDE) can be used to select the region of space with the highest probability:(30)$$x_{0:t}^{MAP}=\underset{x_{0:t}}{\mathrm{argmax}}p_{X|Z}\left(x_{0:t}\left|z_{1:t}\right.\right)\approx \underset{x_{0:t}}{\mathrm{argmax}}\sum_{i=1}^{N_{pts}}K\left({\tilde{w}}_{t}^{\left(i\right)}{\tilde{x}}_{0:t}^{\left(i\right)}\right),$$
where *K* is a 2-D positive kernel function. In our switching observation model particle filter, estimation of the posterior in Equation (Equation 21) from the system evolution models in Equation (Equation 22) can be performed by applying the following steps:Initialize the filter by drawing particles using their prior probability density functions: $x_{0}^{\left(i\right)}∼p_{0}\left(x_{0}\right)$, $\sigma_{0}^{\left(i\right)}∼p_{0}\left(\sigma_{0}\right)$, $a_{0}^{\left(i\right)}∼p_{0}\left(a_{0}\right)$ and set equal weights $w_{0}^{\left(i\right)}\leftarrow 1/N$.For each step t=1,2,... perform sampling according to a proposal function *q* (or the transition model $p_{X|X}\left(x_{t}|x_{t-1},\sigma_{t}\right)$ for bootstrap PF). For each particle sample the sensor state variable ${\tilde{c}}_{t}^{\left(i\right)}∼q_{C|X,A,Z}\left(c_{t}|x_{t-1}^{\left(i\right)},a_{t-1}^{\left(i\right)},z_{t}\right)$, the state vector ${\tilde{x}}_{t}^{\left(i\right)}∼q_{X|X,C,Z}\left(x_{t}|x_{t-1}^{\left(i\right)},{\tilde{c}}_{t}^{\left(i\right)},z_{t}\right)$, the probabilities ${\tilde{a}}_{t}^{\left(i\right)}∼q_{A|A,C,\Sigma }\left(a_{t}|a_{t-1}^{\left(i\right)},{\tilde{c}}_{t}^{\left(i\right)},\sigma_{t-1}^{\left(i\right)}\right)$ and the hyper-parameter vector ${\tilde{\sigma }}_{t}^{\left(i\right)}∼q_{\Sigma |\Sigma ,A}\left(\sigma_{t}|\sigma_{t-1}^{\left(i\right)},{\tilde{a}}_{t}^{\left(i\right)},a_{t-1}^{\left(i\right)}\right)$.Update the weights using the new observation $z_{t}$ using the appropriate observation model, with a slight abuse of notations:
(31)$${\tilde{w}}_{t}^{\left(i\right)}\propto w_{t-1}^{\left(i\right)}\frac{p\left(z_{t}\left|{\tilde{x}}_{t}^{\left(i\right)},{\tilde{c}}_{t}^{\left(i\right)}\right.\right)p\left({\tilde{x}}_{t}^{\left(i\right)}|x_{t-1}^{\left(i\right)}\right)p\left({\tilde{c}}_{t}^{\left(i\right)}|{\tilde{a}}_{t}^{\left(i\right)}\right)p\left({\tilde{a}}_{t}^{\left(i\right)}|a_{t-1}^{\left(i\right)},{\tilde{\sigma }}_{t}^{\left(i\right)}\right)p\left({\tilde{\sigma }}_{t}^{\left(i\right)}|\sigma_{t-1}^{\left(i\right)}\right)}{q\left({\tilde{x}}_{t}^{\left(i\right)}|x_{t-1}^{\left(i\right)},{\tilde{c}}_{t}^{\left(i\right)},z_{t}\right)q\left({\tilde{c}}_{t}^{\left(i\right)}|x_{t-1}^{\left(i\right)},a_{t-1}^{\left(i\right)},z_{t}\right)q\left({\tilde{a}}_{t}^{\left(i\right)}|a_{t-1}^{\left(i\right)}\right)q\left({\tilde{\sigma }}_{t}^{\left(i\right)}|\sigma_{t-1}^{\left(i\right)},{\tilde{a}}_{t}^{\left(i\right)},a_{t-1}^{\left(i\right)}\right)}$$
and normalize the weights such that $\sum_{i=1}^{N}w_{t}^{\left(i\right)}=1$.Compute the effective sample size $N_{eff}$, approximately estimated from $N_{eff}=1/\sum_{i}{\left(w_{t}^{\left(i\right)}\right)}^{2}$ and re-sample when $N_{eff}$ falls to some fraction of the actual samples (say 1/2) in order to avoid the particle impoverishment problem.

The task of the proposal functions *q* is to provide the most probable state space configuration at time *t* given the newly observed data $z_{t}$. For the state vector $x_{t}$ which explains the spatial object characteristics, the optimal proposal function $q_{X|X,C,Z}\left(x_{t}|x_{t-1}^{\left(i\right)},{\tilde{c}}_{t}^{\left(i\right)},z_{t}\right)=p_{X|X,C,Z}\left(x_{t}|x_{t-1}^{\left(i\right)},{\tilde{c}}_{t}^{\left(i\right)},z_{t}\right)$ can be computed by applying the Kalman Filter update step for each particle $x_{t-1}^{\left(i\right)}$ as explained by Van Der Merwe et al. [49]. For the sensor state variable $c_{t}$ Caron et al. [43] approximate the optimal proposal $q_{C|X,A,Z}\left(c_{t}|x_{t-1}^{\left(i\right)},a_{t-1}^{\left(i\right)},z_{t}\right)$ with an extended Kalman filter update step.Lastly, the proposal function for the probabilities $q_{A|A,C,\Sigma }\left(a_{t}|a_{t-1}^{\left(i\right)},{\tilde{c}}_{t}^{\left(i\right)},\sigma_{t-1}^{\left(i\right)}\right)$ can be computed in closed form as given in [43]:$$q_{A|A,C,\Sigma }\left(a_{t}|a_{t-1}^{\left(i\right)},c_{t}^{\left(i\right)},\sigma_{t-1}^{\left(i\right)}\right)=D\left(\left(\sigma_{t}^{\left(i\right)}+1\right)a_{t-1}^{{}^{\prime }\left(i\right)}\right),$$
where the individual components of the vector $a_{t-1}^{{}^{\prime }\left(i\right)}$ are computed as:(32)$$a_{j,t-1}^{{}^{\prime }\left(i\right)}=\left(\frac{\sigma_{t}^{\left(i\right)}}{\sigma_{t}^{\left(i\right)}+1}a_{j,t-1}^{\left(i\right)}+\frac{\delta_{j}\left(c_{t}\right)}{\sigma_{t}^{\left(i\right)}+1}\right),$$
for $j=1,...,n_{c}.$ This Switching Observation Model Particle Filter mechanism can be implemented relatively easily assuming that an observation $z_{t}$ is available to guide particle sampling by proposal functions. However, in real-world multi-target tracking applications this is not always the case. As we previously discussed, multiple factors can cause detections to be missing which impacts the accuracy of the PF and sometimes even compromises its convergence. Therefore, it is of crucial importance to the trackers stability to accurately model missing observations.

### 4.5. Handling Missing Detections

Standard Monte-Carlo Bayesian filters perform sampling using a proposal function based on the new detection whereas weights are updated using Equation (Equation 31). As detections become missing, sampling becomes compromised and weight updates are no longer possible. This is because the proposal functions are conditioned on the new detection. In this subsection we propose to use an adaptation of the multiple imputations particle filter (MIPF) for track updates when detections are missing. Originally introduced in the book by Rubin [50] and later used in the papers [39,41], the MIPF extends the PF algorithm by incorporating a multiple imputation (MI) procedure for cases where measurements are not available so that the algorithm can include the corresponding uncertainty into the estimation process. The main statistical assumption in this approach is that the missing mechanism is missing at random (MAR). This means that the predisposition for a detection to be missing does not depend on the missing detection itself, but can be related to observed ones. For example, when a person being tracked becomes occluded by another person, one detection might not be correctly associated or even not reported at all by the detector. The missing detection is conditioned on the fact that there is a presence of an occluder, so, good techniques for imputing MAR data need to incorporate variables that are related to the missingness. We will show how in our application a detection $\left(z_{j},f_{j}\right)$ which is neither detected or associated to a track $\left(x_{i},g_{i}\right)$ can be approximated by sampling from a likelihood function without association at locations indicated by the prior, i.e., the estimated position $p_{X|Z}\left({\hat{x}}_{t}|z_{0:t-1}\right)$. This way our tracker can update the particles and track multiple hypotheses of a track until we have better evidence to decide which one is right. In the paper [40], Zhang et al. provide more details about the MIPF and prove the almost sure convergence of this filter.

In our case of missing detections, we use the set $\left\{c_{t}^{\left(i\right)}\right\}$ of indicator particles to explain this degenerate operation mode of the fused sensor. For the sake of notational simplicity we will drop the track and detection feature vectors g and f. We will use the auxiliary variable $y_{t}$ to model missing observations which form the partitioned vector $u_{t}=\left(y_{t},z_{t}\right).$ This vector consists of $y_{t}$ which corresponds to the missing part and $z_{t}$ is the observed part of a detection’s ground plane coordinates. The switching observation model Particle Filter algorithm can then be applied irrespective whether u consists of z or y. Thus, depending on the origin of the peak $z_{i,t}$ the observation model can switch between the following states: (33)$$u_{t}=\left\{\begin{array}{cc} {\tilde{y}}_{t} & \mathrm{if}c_{t}=0,\;\left(\mathrm{missing}\mathrm{obseration}\right)\\ x_{t}+w_{1,t};\;w_{1}∼N_{2}\left(0,R_{1}\right) & \mathrm{if}c_{t}=1,\;\left(\mathrm{radar}\mathrm{only}\right)\\ x_{t}+w_{2,t};\;w_{2}∼N_{2}\left(0,R_{2}\right) & \mathrm{if}c_{t}=2,\;\left(\mathrm{camera}\mathrm{only}\right)\\ x_{t}+w_{3,t};\;w_{3}∼N_{2}\left(0,R_{3}\right) & \mathrm{if}c_{t}=3,\;\left(\mathrm{radar}\mathrm{and}\mathrm{camera}\right) \end{array}\right.$$

Using the indicator variables, Equation (Equation 18), $c_{t}$ for the response of the sensor time *t*, the posterior PDF, Equation (Equation 21), can be written as:(34)$$p_{X,C,A|Z}\left(x_{t},c_{t},a_{t}\left|z_{t}\right.\right)=\int p_{X,C,A|Z,Y}\left(x_{t},c_{t},a_{t}\left|z_{t}\right.,y_{t}\right)p_{Y|Z,C}\left(y_{t}\left|z_{t},c_{t}\right.\right)dy,$$
where assuming that the missing mechanism is independent of the missing detections given the observed ones:(35)$$p_{C,A|Y,Z}\left(c_{t},a_{t}\left|y_{t},z_{t}\right.\right)=p_{C,A|Z}\left(c_{t},a_{t}\left|z_{t}\right.\right),$$
using the formulation in [50] we rewrite the posterior as:(36)$$p_{X,C,A|Z}\left(x_{t},c_{t},a_{t}\left|z_{t}\right.\right)=\int p_{X,C,A|Z,Y}\left(x_{t},c_{t},a_{t}\left|z_{t},y_{t}\right.\right)p_{Y,Z}\left(y_{t}\left|z_{t}\right.\right)dy_{t},$$
which means that the statistical model of the missing information is not necessary. In this special case, the posterior distribution as approximated in Equation (Equation 28), can be computed using $N_{imp}$ amount of imputed particles:(37)$$p_{X,C,A|Z}\left(x_{t},c_{t},a_{t}\left|z_{t}\right.\right)=\lim_{N_{imp}\to \infty }\frac{1}{N_{imp}}\sum_{i=1}^{N_{imp}}P_{N}\left(dx_{t},c_{t},a_{t}\left|z_{t},y_{t}^{\left(i\right)}\right.\right),$$
where the multiple imputations $y_{t}^{\left(i\right)}∼p_{Y|Z}\left(y_{t}\left|z_{t}\right.\right)$ are not conditioned on past detections and the state transition equation. We adopt the proposed solution devised in [40] to resolve this deficiency by drawing imputations from the missing data probability density which is unknown, but can be approximated from the posterior. Assuming no detections went missing prior to *t*:(38)$$p_{Y|Z}\left(y_{t}\left|z_{0:t}\right.\right)=\int p_{Y|X,C}\left(y_{t}\left|x_{t},c_{t}\right.\right)p_{X,C,A|Z}\left(x_{t},c_{t},a_{t}\left|z_{0:t}\right.\right)dx_{t},$$

However, in order to get a good estimate of the posterior it is required that detections were present in the time instances leading up to the missing detection. Since we cannot sample directly from the updated posterior $p_{X,C,A|Z}\left(x_{t},c_{t},a_{t}\left|z_{t}\right.\right)$ (due to missing observation: $z_{t}=⊘$) we compute use an approximation by estimating posterior with no regard for missing data using Equation (Equation 28). This means that the particles $x_{t-1}^{\left(i\right)}$ are propagated using the state transition model to obtain an estimated PDF, formed by ${\tilde{x}}_{t}^{\left(i\right)}$ and $w_{t-1}^{\left(i\right)}$. In practice, a missing detection will almost certainly be caused by a localized change of sensor mode $c_{t}$ due to the loss of signal strength, occlusion, ambiguous association or noise. An imputed detection can therefore be simulated using the sensor model and the expected position of the tracked object. At the moment of missing detection, we can let the sensor mode evolution model Equation (Equation 22) choose the most likely course of evolution of $c_{t}$. It is safe to assume that the missing detection PDF is the same as that of the observed data, $p_{Y|X,C}\left(y_{t}\left|{\tilde{x}}_{t}^{\left(i\right)},c_{t}^{\left(i\right)}\right.\right)=p_{Z|X,C}\left(z_{t}\left|{\tilde{x}}_{t}^{\left(i\right)},c_{t}^{\left(i\right)}\right.\right),$ so we can use the imputation proposal function $\phi_{Y|Z}\left(y_{t}\left|,z_{0:t}\right.\right):$(39)$$p_{Y|Z}\left(y_{t}\left|z_{0:t}\right.\right)\approx \phi_{Y|Z}\left(y_{t}\left|,z_{0:t}\right.\right)=\sum_{i=1}^{N_{pts}}{\tilde{w}}_{t}^{\left(i\right)}p_{Y|X,C}\left(y_{t}\left|{\tilde{x}}_{t}^{\left(i\right)},c_{t}^{\left(i\right)}\right.\right),$$
from which we draw imputations $y_{t}^{\left(j\right)}∼\phi_{Y|Z}\left(y_{t}\left|,z_{0:t}\right.\right).$ Practically, (Equation 38) stipulates that the set of simulated detections (imputations) will be generated using the observation models defined by each particle $c_{t}^{\left(i\right)}$. This procedure is clearly illustrated on the two plots on Figure 5. Thus, our definition of the fused sensor model, Equation (Equation 33), for the missing detection case becomes:(40)$$u_{t}={\left\{y_{t}^{\left(j\right)}\right\}}_{N_{imp}}∼\phi_{Y|Z}\left(y_{t}\left|z_{t}\right.\right)\approx \sum_{i=1}^{N_{pts}}{\tilde{w}}_{t}^{\left(i\right)}p_{Y|X,C}\left(y_{t}\left|{\tilde{x}}_{t}^{\left(i\right)},c_{t}^{\left(i\right)}\right.\right).$$

According to Kong et al. [51] we can use these complete data sets, $u_{t}^{\left(j\right)}=\left(y_{t}^{\left(j\right)},z_{t}\right),$ to compute an approximation of the posterior PDF. Substituting $u_{t}$ in Equation (Equation 36) yields:(41)$$p_{X,C,A|Z}\left(x_{t},c_{t},a_{t}\left|z_{1:t}\right.\right)=\int p_{X,C,A|U,Z}\left(x_{t},c_{t},a_{t}\left|u_{0:t-1},z_{t}\right.\right)p_{Y,Z}\left(y_{t}\left|z_{0:t}\right.\right)dy_{t},$$
where the approximate PDF is computed as the Monte Carlo simulation using $N_{imp}$ imputations. Thus, by substituting $u_{t}^{\left(j\right)}$ into Equation (Equation 37) we get:(42)$$p_{X,C,A|Z}\left(x_{t},c_{t},a_{t}\left|z_{t}\right.\right)\approx P_{N}\left(dx_{t},c_{t},a_{t}\left|z_{t}\right.\right)=\lim_{N_{imp}\to \infty }\frac{1}{N_{imp}}\sum_{j=1}^{N_{imp}}P_{N}\left(dx_{t},c_{t},a_{t}\left|u_{0:t-1},u_{t}^{\left(j\right)}\right.\right),$$
which is computed by performing particle filtering treating each imputation $u_{t}^{\left(j\right)}$ as a detection:(43)$$P_{N}\left(dx_{t},c_{t},a_{t}\left|u_{0:t-1},u_{t}^{\left(j\right)}\right.\right)\approx \sum_{i=1}^{N_{pts}}w_{t}^{\left(j,i\right)}\delta_{{\tilde{x}}_{t}^{\left(j,i\right)}}\left(dx_{t}\right),$$
where ${\tilde{x}}_{t}^{\left(j,i\right)}$ is the *i*-th particle for the *j*-th imputation at time *t* and $w_{t}^{\left(j,i\right)}$ is the respective weight estimated from the most likely observation model $c_{t}^{\left(i\right)}$. Finally, by substituting Equation (Equation 43) into Equation (Equation 42) we obtain a form to practically compute an approximation of the posterior PDF when observations are missing and replaced by imputations: (44)$$p_{X,C,A|Z}\left(x_{t},c_{t},a_{t}\left|z_{0:t}\right.\right)\approx \frac{1}{N_{imp}}\sum_{j=1}^{N_{imp}}\sum_{i=1}^{N_{pts}}w_{t}^{\left(j,i\right)}\delta_{{\tilde{x}}_{t}^{\left(j,i\right)}}\left(dx_{t}\right).$$

Two problems arise when applying the multiple imputations PF for real-time application. Firstly, its computation is prohibitively expensive because each time a detection is missing, the particle filter needs to perform $N_{imp}$ updates treating each imputation $u_{t}^{\left(j\right)}$ as a simulated detection and then average the results; double sum in Equation (Equation 44). The complexity lies mainly in the computation of the weights $w_{t}^{\left(i,j\right)}\approx p_{Y|X,C}\left(y_{t}^{\left(j\right)}\left|{\tilde{x}}_{t}^{\left(i\right)},c_{t}^{\left(i\right)}\right.\right)$ which in most cases requires $N_{pts}\times N_{imp}$ evaluations of sensor model. Secondly, since we are dealing with a switching observation model, the accuracy of imputed particles relies on the accuracy of the estimates $\left({\tilde{x}}_{t}^{\left(i\right)},c_{t}^{\left(i\right)}\right)$ which are in turn driven by available detections $z_{0:t-1}$ from the past. In cases when detections are missing in short bursts, updating the evolution model Equation (Equation 24) yields the most likely sensor mode $c_{t}$ which can be accurate enough for drawing imputations and estimating the posterior PDF. However, when detections are missing over an extended time interval, for example, more than a few update cycles, the state transition models can quickly lead to an uninformative vector $a_{t}$, meaning that the states of all sensor mode particles $c_{t}^{\left(i\right)}$ become equally likely. This results in diminished informativeness of the imputations and tracking becomes no better than using motion prediction alone. Thus, without accurate detection information, the tracker is very likely to diverge over time.

Our proposed solution improves the conditioning of imputations on the current sensor data which went missing for various reasons. Instead of sampling from the approximate proposal function in Equation (Equation 40), for each position $\left(\rho_{i},\theta_{i}\right)$ of the posterior estimate ${\tilde{x}}_{t}^{\left(i\right)}$ we compute the likelihood without to the closest detection without association $L_{fused}$. Practically, $L_{fused}$ considers all possible detections $z_{j}$ with no regard to detection thresholds:(45)$$L_{fused}=p_{Z|X}\left(z_{j}|{\left(\rho_{i},\theta_{i}\right)}_{t}\right)\approx \sum_{i=1}^{N_{pts}}{\tilde{w}}_{t}^{\left(i\right)}p_{Y|X,C}\left(y_{t}\left|{\tilde{x}}_{t}^{\left(i\right)},c_{t}^{\left(i\right)}\right.\right);\forall \left(\rho_{i},\theta_{i}\right),\forall z_{j},$$
where it is important to note that we do not use the KL part of the image likelihood in Equation (Equation 16). Using this method, the particle weight update can be performed as:(46)$$w_{t}^{\left(i\right)}\leftarrow \sum_{j=1}w_{t}^{\left(i,j\right)}=\sum_{j=1}w_{t-1}^{\left(i\right)}L_{fused,t}\left(y_{t}^{\left(j\right)}\right),$$
which is better conditioned on the sub-threshold sensor measurements compared to using no sensor measurements at all and thus relying only on the state evolution and sensor models in Equation (Equation 39).

We argue that in the proposed detector-tracker design, the missing part $y_{fused,t}$ of a detection $z_{fused,t}$ often remains hidden bellow a detection threshold or it is discarded due to likelihood gating which safeguards against ambiguous associations. Therefore, by sampling from $L_{fused}$ it is possible to re-use the weak information in regions where a detection is indicated by the posterior $p_{X,C,A|Z}\left({\tilde{x}}_{t},{\tilde{c}}_{t},{\tilde{a}}_{t}\left|z_{t-1}\right.\right)$. In our proposed approach we draw a single $N_{imp}=1$ imputation ${\left\{y_{t}\right\}}_{N_{imp}}$ according to Equation (Equation 45) and compute the weights $w_{t}^{\left(i\right)}$ at locations $y_{t}$, $w_{t}^{\left(i\right)}=w_{t-1}^{\left(i\right)}L_{fused,t}\left(y_{t}\right)$. Using this approach, simulated observations will be drawn from the posterior and their likelihood gets re-evaluated skipping the association algorithm. This approach has the practical implication that the computational load of updating the particle weights is reduced to a single computation per particle at the increased cost of finding for the closest detection in Equation (Equation 45). However finding the closest detection $z_{j}$ to each particle of any track $\tilde{x}{}_{j}^{\left(i\right)}$ and thus the likelihood without association $L_{fused}$ can be performed efficiently by pre-computing this likelihood over a rasterized ground plane grid where each sector can be selected to cover a reasonably large area of equal likelihood. Thus, particles $\tilde{x}{}_{j}^{\left(i\right)}$ falling within the same sector of $L_{fused}$ can share the same likelihood values without loss of information. Since we expect that in a real-world scenario there will be many missing detections, at each time step *t*, we pre-compute the likelihood without association for each spatial location $\left(\rho_{i},\theta_{i}\right)$ over a rasterized grid with range and azimuth density equal with that of the radar sensor. An example of one such grid is shown in the bottom part of the system diagram on Figure 2. Whenever a track update needs to be made without a detection, the tracker can easily sample an imputation using this grid at $\tilde{x}{}_{j}^{\left(i\right)}$ and use the values as a probability mass function. The downside to this approach is that we allow for the aggregated likelihood in $L_{fused}$ to update any unassociated track. This means that, in rare cases, multiple nearby and unassociated tracks can be updated using the same information which will inadvertently lead to tracks converging to each other and merging. However, one can argue that in such situations the limited evidence does not support the existence of more than one track and multiple hypotheses should be merged into one.

### 4.6. Bootstrap Particle Filter

Finding effective proposal functions *q* is problematic since these functions have to approximate the unknown posterior distribution including all its modes as well as tails. When this posterior becomes multi-modal or heavy-tailed, the use of simple parametric proposal functions can lead to ill-informed sampling. In reality, this means that particles will be sampled near a single mode and/or not cover the tails of the actual distribution. It is known that the optimal proposal function, that is, the one minimizing the estimation error, is a multivariate Gaussian formed by applying a Kalman update step on each particle using the current observation. For each particle, a Kalman filter estimates the mean and covariance matrix of the multivariate Gaussian proposal distribution. The particle filter then draws new particles, each from their corresponding proposal distribution. Although this approach is proven to minimize the estimation error, details in [43], it assumes the availability of observation $z_{t}$ at each time step which is not guaranteed. The Kalman update step when data is missing is ill-posed, in a sense that imputations need to be drawn from a state estimate which needs a proposal function conditioned on the missing observation. Even in non-degenerate cases, $c_{t}\neq 0$, running the (Kalman filter) KF update for each particle for multiple tracks is computationally prohibitive. Therefore, we choose to use the bootstrap particle filter [46], which ignores the latest observation during the prediction step. Since the detection might be missing in the current step, the motivation for using the bootstrap PF is sound. The bootstrap PF uses the state and indicator variable evolution models as proposal functions making the particle weight updates in Equation (Equation 31) depend only on the likelihood term:(47)$$w_{t}^{\left(i\right)}=w_{t-1}^{\left(i\right)}p_{Z|X,C}\left(z_{t}\left|{\tilde{x}}_{t}^{\left(i\right)},{\tilde{c}}_{t}^{\left(i\right)}\right.\right).$$

This design choice makes increasingly more sense as the proportion of missing observations increases. Depending of the availability of the detection $z_{t}$, we either update the particle weights using the likelihood of the optimally associated detection Equation (Equation 47); $\left(c_{t}\in \left\{1,2,3\right\}\right)$ or use the likelihood without association to draw imputations and update the particle weights with Equation (Equation 46); $\left(c_{t}≜0\right)$ accordingly. In the latter case, the efficiency of the estimate can be approximated as ${\left(1+\frac{\gamma }{N_{imp}}\right)}^{-\frac{1}{2}}$, expressed in units of standard errors, where $N_{imp}$ is the number of imputations and γ is the fraction of missing information in the estimation, more details in [50]. Finally, after the PF weights are updated the last step of the bootstrap particle filter is to apply importance re-sampling of the set of particles $\left\{x_{t}^{\left(i\right)},w_{t}^{\left(i\right)},c_{t}^{\left(i\right)},a_{t}^{i}\right\}$ to increase the effectiveness of the limited number of particles. Re-sampling can be performed at time intervals controlled by the effective sample size $N_{eff}$ as we explained in Section 4.4.

## 5. Evaluation and Results

### 5.1. Datasets and Metrics

In this section, we present the experimental methodology for assessing the performance of individual components of the proposed system as well as the performance of the complete system. In order to isolate and measure the effects of the individual detector and tracker parts, we used highly realistic simulations, explained in more detail in Section 5.2.1. However, for evaluating the complete system, we are unable to realistically simulate all internal and external factors. Therefore, these experiments were performed using annotated real-world data captured by real sensors in an uncontrolled environment. As of the time of writing this paper the availability of quality camera-radar datasets tailored for autonomous driving is very limited. This scarcity is a consequence of to the novelty of automotive radar technology which many companies consider a closely guarded secret. One of the few public datasets containing, among other sensor modalities, automotive radar data is the recently introduced nuScenes dataset [52]. However, this dataset contains radar data with sparsely populated 2-D radar targets. The very recent public dataset (OLIMP [53]) tailored for VRU detection and tracking incorporates measurements from both camera and ultra wide-band radar sensors. The captured radar data, however, is incomplete as it lacks dense azimuth information which makes it incompatible to our method. The Astyx HIRES2019 dataset [54] contains semi-dense radar data in the form of up to 1000 processed radar targets per frame. Even though this dataset might seem like a solid benchmark for our system, upon closer inspection we found that the majority of the objects are of vehicle targets and not VRUs. The information provided in these datasets is always at the target level with raw radar readouts being lost. Our cooperative fusion algorithm requires a complete range-Doppler-azimuth radar cube to perform accurate, high recall, radar detection making available datasets inadequate. Lastly, the dataset most applicable to our work was captured by the Intelligent Vehicles at TU Delft [55] and offers synchronized camera and raw radar 3D cubes. Unfortunately, due to a non-disclosure agreement issues the complete data is not available at the time of writing and simulations are provided as placeholders. We were therefore motivated to capture, annotate and run experiments on our own dataset, details in Section 5.3. During the capture and annotation of the data we strived to create a lightweight, yet high quality dataset that can stress detection and tracking performance in normal and challenging driving conditions.

The proposed perception system is intended to serve as an input to automated collision avoidance and predictive path planning and therefore, it is of critical importance that the system recalls the most amount of VRUs at the lowest false alarm rate. The position estimates of detected VRUs should also be within an acceptable range of their true position in the real world. At the same time, the system should run on-line while algorithmic complexity needs to allow for a real-time operation using the resources available on an onboard vehicle computer. These contradicting requirements pose a non-trivial parameter optimization problem. In order to count an estimate as a true positive, its position has to be close to the ground truth. However, a state estimate needs to be made as quickly as possible, reducing the latency for emergency braking. Therefore, we focus on measuring the average precision (AP) and multi-object detection/tracking precision (MODP/MOTP) of the VRUs perceived by the system. AP is computed by adjusting the detection/tracking operating point to query the precision at $Q_{REC}$ uniformly spaced recall points:(48)$$AP=\frac{\sum_{i}^{Q_{REC}}PRECISION_{i}}{Q_{REC}},$$
where precision defines the ratio detected relevant objects, true positives (TP), and all detected objects including false positives (FP):(49)$$PRECISION=\frac{TP}{TP+FP}.$$

Recall defines the ratio of relevant objects being detected and all relevant objects:(50)$$RECALL=\frac{TP}{TP+FN},$$
where FN are all relevant objects missed by the detector/tracker. For an object to be considered detected/tracked, the respective algorithm output has to fall within a spatial gate of 1.5 m of the ground truth. Multiple detections falling within this spatial gate are being counted as false positives. AP is an excellent measure for the quality of the detector/tracker, judging the capability to extract objects while minimizing false alarms. Effects such as reduced object recall and increase in false positives, signs of a low signal to noise ratio, will both reduce the AP score.

We use the detection precision metric MODP, to measure the absolute positional accuracy of detected VRUs on the ground plane regardless of AP. In this context, absolute accuracy is computed through the average mean squared error (MSE) for matched object-hypothesis pairs on the ground plane. In the literature average MSE is synonymous with Multiple Object Detection/Tracking Precision (MOTP/MODP) [5] with the general definition:(51)$$MOTP=\frac{\sum_{t,i}d_{t,i}}{\sum_{t}n_{t}},$$
where $d_{t,i}$ measures the distance between a ground truth object $x_{t,i}$ and predicted object position ${\tilde{x}}_{t}^{MAP}$ (in the case of MOTP), or an observation $z_{t}$ (in the case of MODP). The normalizing factor $n_{t}$ counts the total number of detected objects. If we choose to use the squared Euclidean distance $d_{t,i}={∥x_{t,i}-z_{t}∥}_{2}^{2}$ as a metric then MODP becomes equivalent to MSE. Other qualitative aspects of tracked trajectories such as track stability, fragmentation and identity switches (details in [5]) are of lesser importance as we deem that these metrics favor off-line tracking algorithms which have access to future observations and have limited importance in measuring on-line perception performance.

### 5.2. Simulation Experiments

In this sub-section, we assess the performance benefits of the switching observation model and multiple imputation strategies in a series of simulated experiments. For measuring the SOM performance, we track a single simulated object which crosses through areas covered by different sensors. This way we can negate the effects on the performance of the multi-object association and the logic of track existence. The hypothesis is that the SOM filter will switch to the correct sensor model when the tracked object crosses the boundary of an area covered by different sensors. Practically, we will measure how quickly the SOMPF converges to the correct observation model state.

For assessing multiple imputation strategies, we perform destructive testing by increasing the rate of missing observations fed to the tracker. This test is designed to measure the tracker’s capability of handling missing observations in a single target tracking scenario. In this second batch of experiments, we generate noisy detections using a single sensor model which helps to reduce the effects of the above-mentioned on-line sensor model estimation. In this experiment we report the positional accuracy of the tracked object of trackers that do not handle missing detections compared to the MIPF. The hypothesis of this experiment is that as the proportion of missing detections increases, tracking with multiple imputations is more effective than relying only on motion models.

#### 5.2.1. Switching Observation Model

In the first experiment, we simulate a sensor array consisting of a camera and radar with overlapping fields of view. Within this sensing area, we let a simulated VRU follow a stochastic trajectory simulating realistic pedestrian walk. The object moves from an area covered by the radar into an area covered by the camera while for a short period it is also visible by both sensors. Noisy observations are generated following the respective sensor model and fed to a SOMPF tracker. The goal is to analyze the convergence of the switching observation model particle filter explained by the set of equations in Equation (Equation 22). To that end, the object x starts moving perpendicular to the ego vehicle starting from $\rho_{0}$ = 20 m, $\theta_{0}=-60^{\circ }$ with initial velocity magnitude of 1.38 ms^−1^ and velocity orientation of $\theta =90^{\circ }$. The motion of the pedestrian is a stochastic process governed by the state evolution model in Equation (Equation 13) characterized by a change in velocity modeled by Equation (Equation 26). At regular intervals of Δt = 100 ms simulated observations are generated by the array of sensors. Both sensors are oriented in such a way that their combined field of view spans over $\pm 90^{\circ }$ with an overlap over the circular sector of $\pm 15^{\circ }$ in front of the vehicle. The radar also sense the area to the left of the overlapping region $\left[-90^{\circ },15^{\circ }\right]$, while the camera can also sense objects to the right $\left[-15^{\circ },90^{\circ }\right]$. For the radar we use the sensor model from Equation (Equation 2), for the camera Equation (Equation 8) and for the overlapping region Equation (Equation 9) respectively. Specifically, we use the following covariance matrices:(52)$$\begin{array}{cc} R_{1}=\left[\begin{array}{cc} 0.339\rho +0.096 & 0\\ 0 & 0.014^{2} \end{array}\right], & R_{2}=\left[\begin{array}{cc} 0.170 & 0\\ 0 & 0.344^{2} \end{array}\right], \end{array}$$
where ρ is the current range of the object while $R_{3}$ computed as in Equation (Equation 10). The main motivation behind this setup is that we can eliminate external factors such as occlusion, loss of detection and ambiguous association and only measure the time it takes for the SOMPF to converge to the correct mode of operation. Consequently, each particle $c_{t}^{\left(i\right)}$ can switch between the three, d=3, observation models: $c_{t}=j;\;j\in \left\{1,..,d\right\}$ according to the probabilities in Equation (Equation 23). The estimated observation model is then computed as the mode of the particle set: ${\tilde{c}}_{t}=\mathrm{mode}\left(\left\{c_{t}^{\left(i\right)}\right\}\right)$. For the evolution of each $a_{t}^{\left(i\right)}$ we use the model Equation (Equation 20) specified as the Dirichlet process Equation (Equation 24) using the spread parameter $\sigma_{0}^{a}=100.$

We present qualitative results of the simulation as three time extracts shown on Figure 6. The plots on the top show a birds eye view of the situation while the plots on the bottom show the evolution of our belief of the observation model against the ground truth. In order to better visualize the behavior of the estimate ${\tilde{c}}_{t}$, Figure 6 shows the set average ${\tilde{c}}_{t}\approx \sum_{i}^{N_{pts}}c_{t}^{\left(i\right)}$ (blue line) compared against the true state (red line). The posterior estimate $p_{X,C|Z}\left(x_{t},c_{t}|z_{0:t}\right)$ can be visualized through the spread and color of the particle cloud. When the simulated pedestrian crosses a boundary of sensor coverage ($t_{c_{1}\to c_{3}}=7.0$ s and $t_{c_{3}\to c_{2}}=14.7$ s) perturbations caused by the evolution model of $a_{t}$ allow for quick exploration of alternative sensor model solutions. On the plot in the middle of Figure 6 we can observe that the posterior takes a concentrated shape because most $c_{t}^{\left(i\right)}$ converge to the observation model of a fused camera-radar sensor. In this single-target scenario, our tracker shows quick convergence to the correct observation mode, needing an average of 5 updates. We expect that the convergence time in the real world, however, to be also affected by factors such as missing observations and, in the case of MOT, noise from faulty observation to track assignments. However, the results from this simulation show that the proposed SOM adequately selects the correct model without explicit information about the sensor layout.

#### 5.2.2. Multiple Imputations, Sample Size

In this sub-section we analyze several methods for handling of missing detections in a realistic VRU tracking simulation, comparing the MIPF to standard trackers such as Kalman and particle filter. To that end, we simulate MAR detections which we feed to each tracker and plot the error in the estimate against the proportion of missing detections. By definition, the KF and PF do not have a mechanism for handling missing observations, however, in practice it is common to estimate the state using only the state evolution model. Using the appropriate motion model, we can expect that such trackers remain convergent when the ratio of missing observations is relatively low. However, in the context of people tracking assumptions like these can lead to track divergence. As an illustration, consider the fact that people make abrupt changes in their trajectory. If observations are missing during these key moments the tracker is very likely to diverge. We suspect that the performance of the MIPF tracker will depend greatly on the imputation sample size $N_{imp}$ and the ratio of missing observations. In order to control for all other factors, for this experiment we use a single sensor, single object setup. This way we can isolate the effect of missing observations on the state estimates. All trackers use the same observation model parameters. PF and MIPF use the same amount of particles $N_{imp}$ and the same behavioral motion model while the Kalman filter uses a constant acceleration motion model.

We present our findings on the two plots in Figure 7 where on the left plot we show the positional errors of the estimates (in terms of MSE) of various trackers against an increasing proportion of missing detections, while on the right we show the respective average time needed for a single track cycle. The baseline algorithms, whose results are shown with solid markers (blue: KF and red: PF) on the left plot on Figure 7 are computed using complete set of detections. The lines with square markers show the performance of Kalman filter, while lines with circle markers indicate Monte-Carlo trackers. As can be expected, increasing the proportion of missing detections raises the error in the state estimate for all trackers. In shades of orange we show the performance of the MIPF using $N_{imp}=5,\;N_{imp}=50,\;N_{imp}=500$ imputations which relates to 1%,10%,100% of the particle size $N_{pts}$. Each data point represents a mean value from running 1000 simulations with perturbed object trajectories. From the left plot we can conclude with a degree of certainty the following:When there are is no loss of detections, KF performs worse than PF with close to 20% larger MSE. This trend continues with the increase of missing detections.When detections are missing, using very few imputations ($N_{imp}=\frac{N_{pts}}{100}$) makes the MIPF estimate worse than predicting the position solely on the state evolution model. This trend is consistent for all evaluated rates of missing observations.When using a larger number of imputations ($N_{imp}\geq 50$) the MIPF estimate is more accurate than the standard PF. This trend is also consistent for every rate of missing observations.

We measured that the MSE of the KF estimate becomes worse than the original detections when more than 20% of the data is missing. For the standard PF filter this point is reached at 47% data loss, while for the MIPF at 52%. The former finding confirms the limited capacity for handling non-linear motion of KF, while the latter indicates the possible benefits of using imputation theory in VRU tracking. Even though we empirically optimized the Kalman filter parameters, the measured MSE of the estimate is consistently worse than the PF. We suspect that this is due to the highly non-linear motion model used to simulate the state transition, that is, the simulated object can sometimes make sudden random changes in their motion. Regardless, when no data is missing, all algorithms produce an estimate which is more accurate than the measurement, dashed line Figure 7. Thus the choice of an appropriate tracker can be made by considering other factors such as algorithmic complexity and memory requirements.

On the right plot on Figure 7 we see the execution time of the three considered trackers. These graphs were generated by fixing the missing detection rate to 25% and the imputation size of $N_{imp}=50.$ For the PF and MIPF we vary the particle set size $N_{pts}$ from 0 to 1000. On the vertical axis, we report the average execution time of a single prediction-update cycle. All three trackers were implemented in MATLAB using optimizations such as vectorization and built-in random generators. The code runs in a Ubuntu 18.04 environment using an Intel Core i7-4930K with 64 GB memory. As expected, the time needed for a PF prediction and update increases with the increase of $N_{pts}$. This trend is also even more evident for the MIPF, where the rate of increase is proportional to the imputation set size $N_{imp}$. We can see from the plot that all three algorithms can be employed in real-time tracking systems, however, design considerations need to be taken when tracking multiple targets. For example, our prototype system uses a sensor readout of 10 Hz which limits the maximum number of targets at 1230 for KF, and 237 for PF and 17 for MIPF (with $N_{pts}=512$).

### 5.3. Real-World Experiments

In this sub-section, we present the methodology and results of our experiments using real-world data. In the first experiment, the detection performance of the proposed cooperative fusion detector is evaluated against an intermediate-fusion and single sensor camera-only detector. In the second experiment, the tracking performance of the proposed tracker is compared against a Kalman filter, a PF, a PF with switching observation models (SOMPF) and a multiple imputations SOMPF. Formally, we will evaluate the validity of following claims:**Detection:** the proposed cooperative camera-radar detector outperforms intermediate-fusion and single sensor detection in both detection quality and detection accuracy.**Tracking:** when the detector fails to detect objects, temporal tracking using the proposed tracker outperforms optimal techniques such as KF and PF, SOMPF, and MI-SOMPF.

Our evaluation dataset contains data captured using an electric tricycle equipped with a sensor array and capturing hardware. The sensor array consists of a forward-looking, wide field of view RGB camera, (GoPro Hero4 black), providing high quality, time-stamped video information, a 77 GHz FMCW radar (Texas Instruments AWR1243) and a 3D lidar (Velodyne VLP-16). The camera was set to record frames in a FullHD resolution at 30 FPS. Its horizontal field of view after accounting for the fish-eye image deformation is approximately $90^{\circ }$. The Velodyne lidar contains 16 laser diodes that measure distance in a 360-degree circle up to 100 m. The beams are uniformly arranged over the $\pm 15^{\circ }$ elevation, while the azimuth resolution is ∼$0.2^{\circ }$. The accuracy of the measured range of each beam is ±3 cm. The AWR1243 radar, on the other hand, contains only 3 transmit and 4 receive antennas. We used a configuration of the AWR1243 radar that allows for a range resolution of 36.5 cm with maximal range of 46.72 m, and Doppler resolution of 0.21 ms^−1^ with a maximal unambiguous velocity of 50 kmh^−1^. Due to the low number of antennas which are arranged in a horizontal pattern, the radar can sense in a single elevation plane with an angular resolution of $11.25^{\circ }$. The sensor constellation is as depicted on Figure 3 with the area of interest defined by the intersection of the camera and radar fields of view ∼$\left[-45^{\circ },\;45^{\circ }\right]$. Vehicle odometry is estimated at run-time using the lidar data and the algorithm from our previous work [56] allowing tracking to be performed accurately in global coordinates.

We selected 16 short and highly representative sequences in which we manually labeled the positions of all VRUs. We use the camera and radar data as input, while the lidar data was used for annotating ground truth VRUs positions. Four annotators were trained and asked to match people visible in the RGB image to objects in the lidar point cloud. The matched objects were ranged, forming labels containing the 2-D ground plane position of each person relative to the sensor array. This manually labeled dataset consists of 1840 annotated frames with 4988 VRU instances. The data covers situations from poorly lit environments (20% of the data) to well-lit sequences captured in daylight. A total of 148 unique VRUs, pedestrians, and cyclists were labeled with each test sequence containing at least one person. On average, there are 9 unique VRUs, with 34 occurrences per sequence. Due to the limited resolution of the available VLP-16 lidar, we were able to only confirm the presence of VRUs within a range of 20 m. Thus all performance numbers are computed for detected and tracked objects within this range while the areas beyond 20m are considered as “don’t care” regions where we ignore detections.

#### 5.3.1. Detection Fusion

To test the detection hypothesis we used the outputs from individual camera and radar object detection CNNs. For the camera object detector, we used the PyTorch implementation (Code available at: https://github.com/eriklindernoren/PyTorch-YOLOv3) of YoLoV3 [8] CNN. The specific model we used was trained on 80 object categories in the MS-COCO dataset [57] where we only selected the output for the class person, since this way we can detect most VRU classes as defined in the introduction. For processing the radar signal, a VRU detection CNN trained to detect micro-Doppler patterns of human body in motion was used [7]. As seen in the introductory diagram Figure 1, the output of this CNN is a 2-D grid of detection scores predicting the position of people. Our first control algorithm is a camera-only detector, where the estimated ground plane position of detected bounding boxes in the image are computed using the back-projection method [34] assuming an average person height of 1.65 m. Our second control algorithm is an intermediate fusion technique for matching camera and radar detections on the image plane explained in Equations (Equation 2)–(Equation 11). The proposed cooperative fusion detector additionally applies the confidence-boosting technique in Equation (Equation 12) where we use the weak detection threshold $\tau_{det}=0.8$ and the control parameter β=0.2. For obtaining the fused detections on the ground plane we used the ground plane sensor model parameters in Equation (Equation 52) and for the image plane matching we compute the Bhattacharyya coefficient using the transformations in Equation (Equation 4) for $R_{1,IP}$ and the following parameters for Equation (Equation 5):(53)$$R_{2,IP}=d_{BB}\left[\begin{array}{cc} 0.0469^{2} & 0\\ 0 & 0.0032^{2} \end{array}\right],$$
where the covariance matrix is relative to the bounding box diagonal $d_{BB}$. These values were learned off-line using the ground truth annotations in the KITTI dataset.

All detected VRUs within 1.5 m to a ground truth object on the ground plane is counted as a true positive, while other detections outside of don’t care regions are treated as false positives. We report the results in Table 1 where we additionally break down the results for each individual sequence comparing AP and MODP scores for the three tested detectors. Set averages, presented on the bottom of the table, and visualized on Figure 8, measure the performance for the low-light and daylight segments as well as for the complete dataset.

In terms of AP, the proposed cooperative fusion outperforms camera only detection by 3.7% and radar-camera early fusion by 3.3%. The performance difference is especially pronounced in low-light sequences where the proposed method shows as much as 32.3% and 30% improvements over the controls respectively. This improvement is a direct consequence of the radar-camera feedback loop used to boost detection confidences, Equation (Equation 12), which is effective in low-light where camera detection performance weakens. The proposed method has increased maximal recall and precision meaning more VRUs are being detected while at the same time producing less clutter. Otherwise, in well-lit sequences, the proposed cooperative fusion method produces slightly better results than intermediate fusion which makes it robust and predictable.

In terms of positional detection accuracy, MODP, we observe a similar trend. Regardless of the number of recalled objects, Equation (Equation 51), camera-radar fusion methods significantly outperform camera-only detection. In low light sequences, intermediate fusion detection reduces the positional error by 0.154 m over camera-only with cooperative fusion improving by an additional 0.008 m. This result means that when fusing radar and camera, detected objects $z_{j}$ are more accurate, while the additional confidence-boosting further improves positional accuracy. In daylight sequences, the differences between the methods are not as pronounced, however, the proposed cooperative fusion method significantly outperforms camera-only detection and is on par with intermediate fusion.

#### 5.3.2. Multi-Object Tracking

For testing the effectiveness of tracking with missing detections, we evaluated the performance of the proposed tracker to several optimal trackers: Kalman filter (KF), standard particle filter (PF), switching observation model particle filter [43] (SOMPF) and a multiple imputations SOMPF with 150 imputations. Each tracker was extended to multiple object tracking using the same track maintenance logic. The KF uses a constant acceleration motion model empirically optimized to match the non-linear models employed by the Monte-Carlo trackers. The number of particles for all experiments was kept the same at $N_{pts}=1000$. The KF and PF use the single observation model from Equation (Equation 9) while the SOM PF, MI-SOM PF, and the proposed trackers switch between three observation models as previously explained. Particle re-sampling for the Monte-Carlo trackers is performed whenever the effective sample size of a track becomes lower than 20% of the number of particles: $N_{eff}\leq \frac{1}{5N_{pts}}$. Finally, we took special care to reset pseudo-random generator seed to the same initial value between all experiments.

We performed two experiments: firstly, by feeding the trackers with detections from the cooperative fusion detector and secondly, by feeding the trackers with ground truth detections using a simulated MAR mechanism. The first experiment is designed to measure the expected tracking performance in a deployment-ready setup where we cannot control for external factors, while the second experiment performs destructive testing and is designed to measure the theoretical capacity for handling missing detections. With these tests we want to investigate the degradation of tracking performance with the worsening of input detections, firstly by using realistic detections with MAR mechanism built-in by design of the detection CNNs, and secondly by controlling the MAR mechanism using an increasing rate of missing detections. We expect that the proposed tracker will be more immune to missing observations as it uses imputations sampled from a proposal function which is better conditioned on the real detections. To that end, for the second experiment, we form a sub-set of camera observations that simulates detections missing at random with probability $p\left(z=\right)=p_{FN}^{\prime }$. By adjusting this probability of a missing detection in the range of $p_{FN}^{\prime }\in \left[0,1\right)$ we perform destructive testing evaluated tracking algorithms, feeding them with temporally inconsistent observations.

We present the findings of our real-world VRU tracking experiment on the bar plot on Figure 9. The height of the gray bars indicates the average precision of the detection input, while the height of the colored bars indicates the average precision of the tracking output for the evaluated trackers. Additionally, we split our test set into sequences with simple and complex tracking situations based on the amount of VRUs present in the scene and the degree of tracking ambiguity caused by occlusion. The “simple” sub-set consists of the sequences $\left(1,2,3,7,8,9,10,11,12,13\right)$ with 1150 frames and 1479 VRUs, while the “complex” sub-set consists of the remaining 6 sequences with 690 frames and 3509 VRUs. From the data visualized on Figure 9 we can conclude that all trackers perform significantly better in simple tracking situations than in complex ones which is also indicated by better object detection at the input. In simple situations (left bar cluster on Figure 9), the best performing algorithm is the SOM-PF which uses state prediction based on motion model alone when detections are missing. In complex scenarios (middle bar cluster on Figure 9) however, the proposed tracker outperforms all other tracking algorithms. The same conclusion can be made when if we average the results for all sequences of the dataset (right bar cluster on Figure 9), where the proposed algorithm performs best, followed by the multiple imputations PF, then the other Monte-Carlo trackers and finally KF. This result confirms our hypothesis that the proposed tracker outperforms the optimal trackers under realistic, less than ideal, object detection.

We present the findings of our destructive testing using ground truth detections with simulated MAR mechanism on Table 2. In each trial we increase the proportion of detections missing at random by 10% and measure the tracking performance for the above-mentioned trackers in both simple and complex sequences. In this second set of experiments we observed the same pattern as in the real world tests: tracking in simple situations achieves significantly higher AP than in complex situations. This trend is true for all levels of missing detections. Additionally, we can conclude that all evaluated trackers demonstrate relatively good performance as long as more than half of the data is not missing. However, the Monte-Carlo trackers outperform the KF in almost all but the trivial tests which shows their increased capacity for tracking the unpredictable motion of VRUs. As can be seen in Table 2 the proposed tracker outperforms all other trackers when the amount of missing detections is between 20% and 80% in simple tracking situations and between 30% and 90% in complex tracking situations. We note that the proposed cooperative radar-camera fusion detector achieves an AP of 63.2% in simple and 57.2% in complex situations which sits in the middle of the simulated MAR rate ranges of this experiment.

These findings indicate that the combination of the proposed cooperative fusion for object detection and the proposed tracker is capable of tracking VRUs more confidently creating a better basis for safer collision avoidance. In a concluding remark, we note that in real-world tracking situations the main benefit of temporal tracking is not an increase in the positional estimate accuracy, but rather an increase in object recall and reduced clutter which greatly simplifies long term path planning as well as collision avoidance algorithms.

## 6. Conclusions

This paper proposes a theoretical framework for accurate tracking of multiple VRUs from multiple, imperfect, sensors in the context of automotive perception. Additionally, we propose various design choices that adapt both the detection and the tracking for real-time operation. The output of the system is a list of confidently tracked VRUs which can be communicated to other perception sub-systems such as collision avoidance or path planning. Our system has two-fold benefits over traditional MOT: firstly, a decision about each VRU position and velocity is made on-line and with no delays, and secondly, the positions and velocities are estimated without discarding any detection information.

Our first contribution is on the detector side, where we propose a cooperative fusion of radar and camera detections by probabilistic matching on the image plane using the sensor model uncertainties. We apply confidence-boosting of detection scores based on the Bhattacharyya coefficient in order to reinforce the uncertain detections of one sensor in close proximity to strong detections of the other sensor. Results from real-world experiments confirm the benefit of our cooperative fusion method in terms of better recall, lower false positives, and higher positional accuracy. The proposed detector outperforms standard intermediate fusion and significantly outperforms camera-only detection. Differences are most evident in low-light sequences where the radar’s luminance invariance recovers information lost in the camera image processing.

A secondary contribution, made to the tracker update logic, facilitates better handling of missing detections which are often encountered in real-world applications. Tracking is performed on the ground plane, agnostic to the sensor configuration, where we assume that individual sensors provide detections that are missing at random. Through the use of a switching observation model, the proposed tracker is able to adapt to changing sensor configurations as well as failures of individual sensors. The novelty lies in degenerate track update cases when detections are missing, where we propose to approximates the missing data using imputation theory. We addressed the conditioning and time complexity problems of the standard MI algorithm by introducing a novel imputation proposal function which uses all detections without association to update particle weights of unassociated tracks. This way we are able to simplify the complex MI particle weight updates enabling our tracker to track multiple objects in real-time. Simulated as well as real-world experiments show that this method has clear advantages over optimal trackers such as KF and PF which by definition rely on an uninterrupted stream of observations. The output of the proposed tracker remains invariant to missing observations even when as many as 50% of the detections are missing at random.

It remains to be seen, however, if the proposed detection and tracking algorithms perform as well in a wider context of sensor configurations. Of special interest for our further study is evaluating the system in surveillance applications where multiple radar-camera arrays observe both joint and disjoint areas. We anticipate that in these situations the switching observation model should also incorporate a switching appearance model that can handle the different object appearance and model sensor to sensor hand-off. Additionally, in surveillance areas of no sensor coverage, the benefit of the proposed multiple imputations sampling function will be diminished. However, the expected position of a tracked person across multiple sensor arrays can be used as a strong detection prior which can easily be exploited in our cooperative fusion framework. Our further plans for the development and implementation of the proposed solution include evaluation of the system in a much wider set of traffic scenarios which will highlight any critical corner cases not present in the current analysis. We are currently in the process of capturing and labeling a larger traffic dataset by an order of magnitude of the one being used in this paper. The main weakness of the presented approach lies in the standardized evaluation protocol and its inability to guarantee a certain safety level in the real world. Although we use standard metrics that are widely used in the detection and tracking literature, we are also working on developing a novel performance metric that more accurately measures the real implication of miss-detected VRUs on safety.

## Figures and Tables

**Figure 1 sensors-20-04817-f001:**
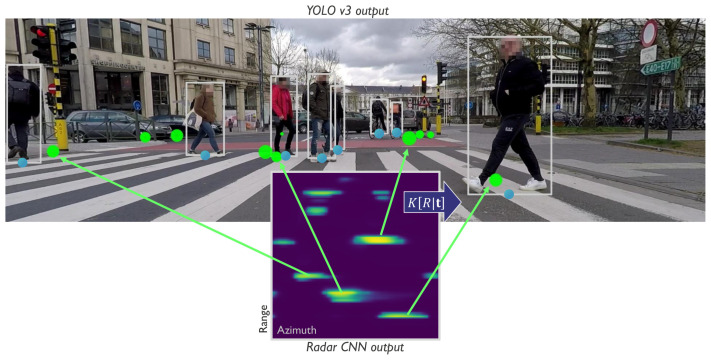
Diagram of the proposed camera-radar detection fusion: detections from a radar CNN are projected using the 3×4 projection matrix $K\left[R|t\right]$ from Euclidean 3-space to an image (green circles) where they are probabilistically matched to image detections (blue circles) using models for the sensor positional uncertainties. The output is a set of fused detections and their image and radar features $\left\{\left(z_{fused,j},f_{fused,j}\right)\right\}.$

**Figure 2 sensors-20-04817-f002:**
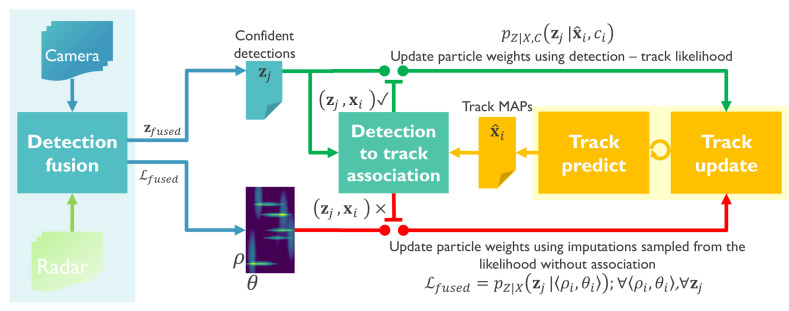
Simplified diagram of the proposed tracking update logic. Depending on the outcome of the detection— track association, track updates are performed either by using association likelihood between a confident detection and a specific track (green path), or using the likelihood for imputing a detection without association (red path).

**Figure 3 sensors-20-04817-f003:**
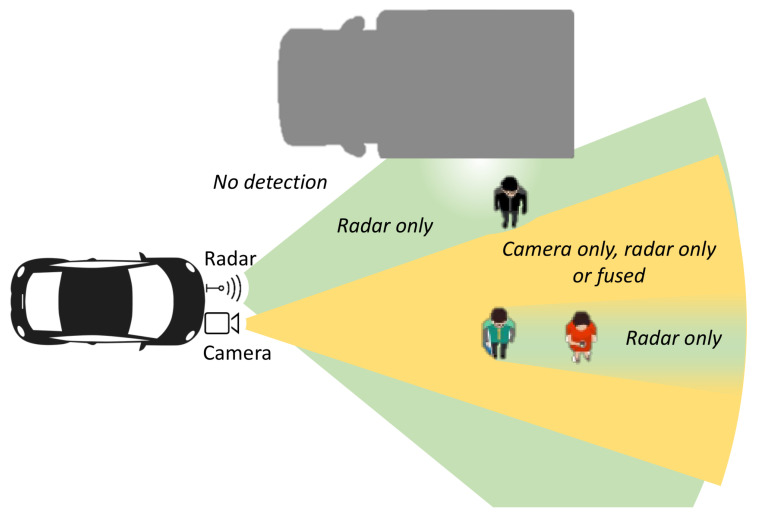
Diagram of the sensor layout showing the different modes of operation of a camera-radar sensor pair. Even in areas covered by both sensors the mode of operation can switch between radar-only, camera-only or combined camera-radar detection depending on the scene layout and characteristics.

**Figure 4 sensors-20-04817-f004:**
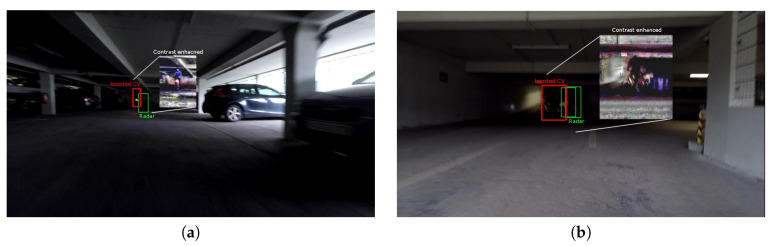
Typical examples of cooperative sensor fusion in poorly lit bicycle/car parking facility. Objects are detected in both modalities and Radar information is used to boost the confidence of computer vision detections. (**a**) Person on a bicycle moving toward the ego vehicle. Initial detection score $s_{cam}=0.221$, after boosting $s_{cam}=0.618$; (**b**) Person on a bicycle moving away from the ego vehicle. Initial detection score $s_{cam}=0.105$, after boosting $s_{cam}=0.535$.

**Figure 5 sensors-20-04817-f005:**
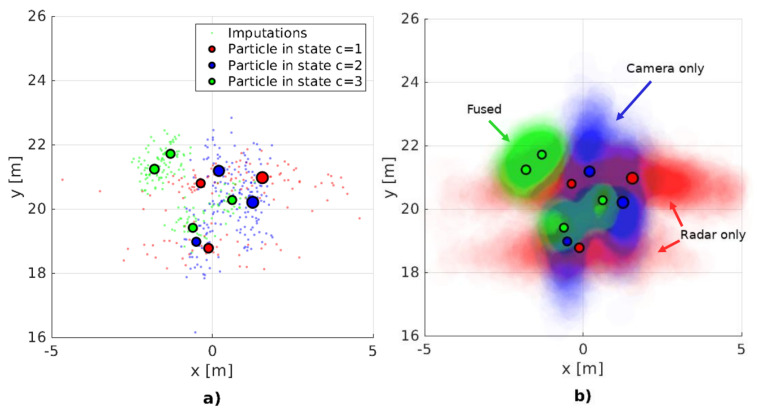
Visualization of the set of particles and imputations drawn by a standard MIPF for a single VRU $x_{t}$ at range of 20 m. On the plot (**a**) one realization of sampling 500 imputations from particles whose colors indicate their respective sensor state variable $c_{t}^{\left(i\right)}$. On the plot (**b**) the underlying imputation proposal function.

**Figure 6 sensors-20-04817-f006:**
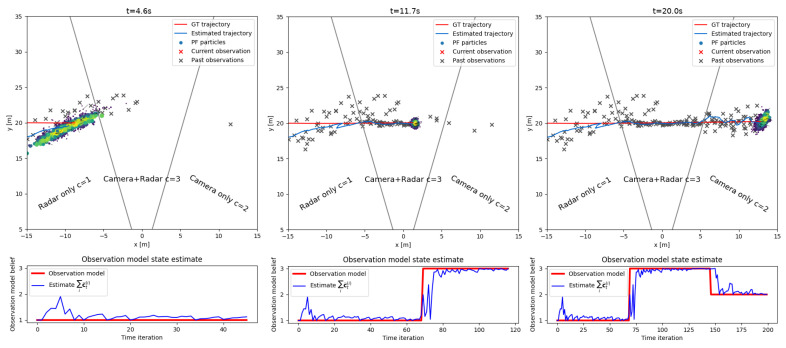
Tracking a simulated target with the switching observation model PF at 100 ms time intervals. On the top we present a situational layout, while on the bottom we show the evolution of the indicator variable $\sum_{i}{\tilde{c}}_{t}^{\left(i\right)}.$ Switching of the underlying sensor model happens at $t_{c_{1}\to c_{3}}=7.0$ s and $t_{c_{3}\to c_{2}}=14.7$ s while the model estimate switches to the correct state at times ${\tilde{t}}_{c_{1}\to c_{3}}=7.6s$ and ${\tilde{t}}_{c_{1}\to c_{3}}=15.5$ s.

**Figure 7 sensors-20-04817-f007:**
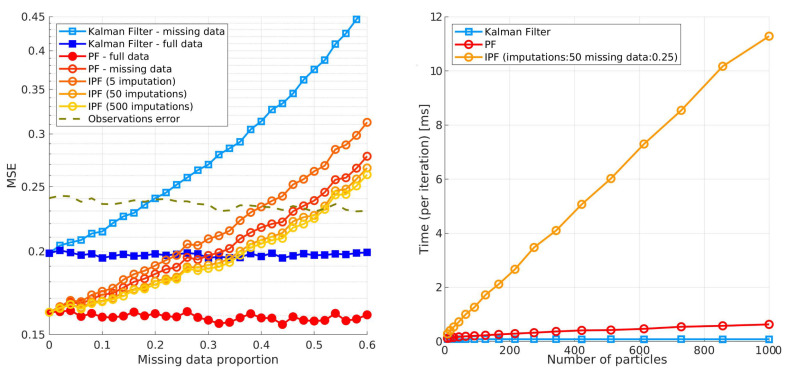
Performance evaluation for tracking a simulated VRU. On the left plot, we show the MSE for track estimates of several trackers at various missing data rates. On the right plot we show the average execution time for a single prediction-update cycle.

**Figure 8 sensors-20-04817-f008:**
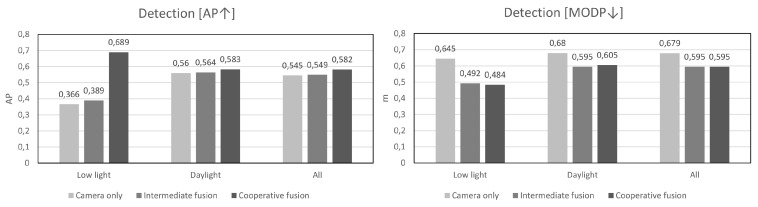
Comparison of VRU detection performance of the proposed cooperative fusion and two control detection algorithms. **Left**: classification performance expressed as Average Precision (higher is better), **Right**: positional accuracy expressed as MODP with Euclidean distance metric (lower is better).

**Figure 9 sensors-20-04817-f009:**
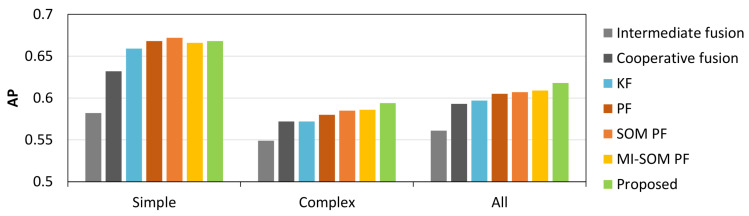
VRU detection (gray bars) and tracking (colored bars) performance expressed as average precision.

**Table 1 sensors-20-04817-t001:** Summary of the dataset and results for a single sensor (camera) and multi-sensor (camera-radar) VRU detection. In terms of AP, the proposed cooperative fusion detector significantly outperforms both other methods in low light and daylight sequences. In terms of MODP, the proposed detector outperforms both methods in low light sequences and has better overall precision. The best performing method is indicated with bold numbers.

Dataset Information			Detection [AP↑]			Detection [MODP↓]		
Sequence	# Frames	# VRUs	Camera Only	Inter-Mediate Fusion	Coop. Fusion	Camera Only	Inter-Mediate Fusion	Coop. Fusion
01	175	55	0.177	0.177	**0.397**	0.753	0.585	**0.529**
02	205	190	0.423	0.448	**0.778**	0.630	0.480	**0.477**
03	345	443	**0.704**	0.694	0.694	0.607	**0.486**	**0.486**
04	125	530	0.632	0.681	**0.695**	0.821	**0.599**	0.602
05	145	705	0.515	0.599	**0.612**	0.699	**0.549**	0.559
06	205	1319	0.606	0.588	**0.609**	**0.626**	0.627	0.637
07	125	290	0.420	0.434	**0.454**	0.782	0.630	**0.629**
08	25	52	**0.905**	0.818	0.818	**0.557**	0.594	0.594
09	55	114	**0.956**	**0.956**	0.955	**0.508**	0.550	0.555
10	85	147	0.424	0.457	**0.474**	0.960	**0.612**	0.617
11	25	10	**1.0**	**1.0**	**1.0**	0.489	**0.359**	**0.359**
12	55	65	**0.802**	0.692	0.692	0.520	**0.444**	**0.444**
13	55	113	0.852	0.887	**0.897**	**0.572**	0.582	0.580
14	105	373	0.246	0.276	**0.293**	0.810	**0.741**	0.746
15	65	330	**0.595**	0.543	0.572	0.746	0.617	**0.621**
16	45	252	**0.477**	0.440	0.445	**0.676**	0.710	0.714
Low light	380	245	0.366	0.389	**0.689**	0.645	0.492	**0.484**
Daylight	1460	4743	0.560	0.564	**0.583**	0.680	**0.599**	0.605
All	1840	4988	0.545	0.549	**0.582**	0.679	**0.595**	**0.595**

**Table 2 sensors-20-04817-t002:** Summary of the capability for handling missing detections of the analyzed tracking algorithms. Each tracker is fed by a reduced sub-set of ground truth detections. The best performing method is indicated with bold numbers.

Input	Tracking Output (Simple) [AP% ↑]					Tracking Output (Complex) [AP% ↑]				
Detections MAR [%]	KF	PF	SOM PF	MI-SOM PF	Pro-Posed	KF	PF	SOM PF	MI-SOM PF	Pro-Posed
10%	**93.02**	90.75	92.39	92.27	92.22	**92.08**	86.5	88.57	88.96	88.99
20%	88.65	89.11	90.4	90.28	**90.56**	**87.32**	83.62	86.26	86.7	86.65
30%	83.31	86.79	88.43	87.55	**88.85**	80.92	79.97	83.16	83.56	**83.85**
40%	76.58	83.67	85.23	84.59	**86.12**	72.17	74.82	76.07	79.69	**79.76**
50%	65.23	77.17	79.09	78.81	**80.19**	61.53	67.81	71.34	71.35	**72.61**
60%	51	65.61	67.88	68.34	**70.73**	48.26	57.84	60.79	60.85	**63.59**
70%	34.13	50.84	53.45	52.73	**55.94**	33.08	44.55	45.5	44.93	**48.45**
80%	18.93	29.32	31.5	30.43	**33.37**	18.16	26.34	26.68	26.85	**29.46**
90%	10.96	19.66	**20.79**	19.24	20.66	10.39	16.17	15.99	16.2	**17.74**

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
