# Peer review of "Cooperative Multi-Sensor Tracking of Vulnerable Road Users in the Presence of Missing Detections"

_sensors, 2020, doi:10.3390/s20174817_

Round 1
Reviewer 1 Report
In the reviewed paper Authors presented a VRU tracking algorithm capable of handling noisy and missing detections from heterogeneous sensors. Authors propose a cooperative fusion algorithm for matching and reinforcing of radar and camera detections using their proximity and positional uncertainty. Interesting paper whose subject matter relates to the current problem. The presented content is clearly described. In my opinion, the paper can be published after taking into account the following remarks:
- all abbreviations, acronyms used in the paper text for the first time should be explained (f.ex. Nimp, KF, PDF. etc.),
- the title of the paper is: "Cooperative Multi-Sensor Tracking of Vulnerable Road Users in the Presence of Missing Detections", but Authors don't expalin in the paper text which kind of road participants is taking into account as a vulnerable road users. Authors should prepare a short paragraph, in which they explain it and refer to the research works dealing with pedestrian traffic (f.ex. Gavrila D.M. (2000) Pedestrian Detection from a Moving Vehicle. In: Vernon D. (eds) Computer Vision — ECCV 2000. ECCV 2000. Lecture Notes in Computer Science, vol 1843. Springer, Berlin, Heidelberg) and bicycle traffic (f.ex. Macioszek E., Świerk P., Kurek A.: The Bike-Sharing System as an Element of Enhancing Sustainable Mobility - A Case Study based on a City in Poland. Sustainability 2020, 12, 3285; doi:10.3390/su12083285). The best place for this paragraph will be Introduction section,
- at the end of section "1. Introduction" The Authors describe what is contained in subsequent chapters, calling them f.ex. "§2". A better definition instead of "§" would be the word "section". It should be replaced,
- on the Figure 1 is used acronym "K[R]t", but their meaning is not explained in the paper text,
- Authors in a few places of the paper wrote what was the purpose of the paper and what the paper contains. This should be standardized in one place (the best place is in introduction section at the end) and not repeated in the paper text few times,
- description of Figure 5 is "Figure 5. Visualization of the set of particles and imputations drawn by a standard MIPF for a single VRU xt at range of 20m. On the plot a) we show one realization of sampling 500 imputations from particles whose colors indicate their respective sensor state variable c(i) t . On the plot b) we show the
underlying imputation proposal function." but it should be write without word "we", it should be written in an impersonal form: "Figure 5. Visualization of the set of particles and imputations drawn by a standard MIPF for a single VRU xt at range of 20m. On the plot a) one realization of sampling 500 imputations from particles whose colors indicate their respective sensor state variable c(i)t .; b) the underlying imputation proposal function.", - in the "Conclusion" section, Authors should also add information about further plans for the development and implementation of the proposed solution as well as its advantages and disadvantages,
- Authors use a personal style when writing a paper, f.ex.: line 1: ..."In this paper we present a..."; or line 2: ..."We propose a cooperative fusion"...; or line 10: ..."we bypass the problem of imputing"... and generally in paper text. It is assumed that the papers are written in an impersonal style, f.ex.: "as adopted", "made", "designed", etc.
Reviewer 2 Report
Dear Authors,
Congratulations on interesting research.
- Please spell-check the manuscript, as there are minor mishaps, and cut sentences, like in the caption of figure 9.
- Please relate your work to real-life systems, such as used on board Tesla vehicles
- Please state the metrological parameters of equipment used (sensors, reference measurements of actual target positions)
Sincerely,
Reviewer
Round 2
Reviewer 1 Report
Authors improved paper according to reviewer comments. Paper can be published in the present form.